# Medium spiny neurons activity reveals the discrete segregation of mouse dorsal striatum

Javier Alegre-Cortés[†], María Sáez[†], Roberto Montanari, Ramon Reig*

Instituto de Neurociencias CSIC-UMH, San Juan de Alicante, Spain

**Abstract** Behavioral studies differentiate the rodent dorsal striatum (DS) into lateral and medial regions; however, anatomical evidence suggests that it is a unified structure. To understand striatal dynamics and basal ganglia functions, it is essential to clarify the circuitry that supports this behavioral-based segregation. Here, we show that the mouse DS is made of two non-overlapping functional circuits divided by a boundary. Combining in vivo optopatch-clamp and extracellular recordings of spontaneous and evoked sensory activity, we demonstrate different coupling of lateral and medial striatum to the cortex together with an independent integration of the spontaneous activity, due to particular corticostriatal connectivity and local attributes of each region. Additionally, we show differences in slow and fast oscillations and in the electrophysiological properties between striatonigral and striatopallidal neurons. In summary, these results demonstrate that the rodent DS is segregated in two neuronal circuits, in homology with the caudate and putamen nuclei of primates.

*For correspondence:
ramon.reig@umh.es

[†]These authors contributed equally to this work

**Competing interests:** The authors declare that no competing interests exist.

## Introduction

The basal ganglia are a group of subcortical nuclei involved in a diversity of functions including motor control, learning, decision making, and reward (*Packard and Knowlton, 2002*; *Schultz et al., 1997*; *Yin and Knowlton, 2006*). The striatum is the main input structure of the basal ganglia, receiving glutamatergic transmission from the cortex and the thalamus (*Alloway et al., 2009*; *Kincaid et al., 1998*; *Wilson, 1987*). In primates and other mammals, the dorsal part of the striatum is formed by the nuclei caudate and putamen, which are anatomically separated by the internal capsule. In contrast, rodent dorsal striatum (DS) is considered a homogeneous structure due to the absence of anatomical border and the copious presence of overlapping axonal connections (*Alloway et al., 2006*; *Hooks et al., 2018*; *Hoover et al., 2003*). However, some recent studies have divided the DS in two regions, dorsomedial (DMS) and dorsolateral (DLS) striatum, based on their behavioral roles (*Graybiel, 2008*; *Hauber and Schmidt, 1994*). While both regions participate in motor control, DLS is often related with habit, stimulus-response associations and navigation, whereas the DMS mediates action-outcome associations, goal-directed actions and flexible shifting between behavioral strategies, suggesting a role in higher cognitive functions (*Faure et al., 2005*; *Hilário and Costa, 2008*; *Lerner et al., 2015*; *Thorn et al., 2010*). In order to understand the basal ganglia circuits and their related behaviors, an essential question should be addressed: Is the functional segregation of the DS supported by two different circuits?

Cortex and thalamus project to the striatum, forming organized glutamatergic synapses along its mediolateral axis, and defining multiple striatal subregions (*Hintiryan et al., 2016*; *Hunnicutt et al., 2016*). This corticostriatal axonal innervation presents a high degree of convergence and divergence (*Flaherty and Graybiel, 1991*) and originates in both hemispheres from different subtypes of pyramidal neurons (*Cowan and Wilson, 1994*; *Hooks et al., 2018*; *Levesque et al., 1996*; *Reiner et al., 2003*; *Wilson, 1987*). Corticostriatal connections innervate both striatonigral and striatopallidal

medium-sized spiny neurons (direct and indirect MSNs, respectively) (*Doig et al., 2010*; *Wall et al., 2013*), which represent 95% of the neurons in the striatum (*Kita and Kitai, 1988*), and different types of interneurons (*Tepper et al., 2008*). The lateral region of the striatum is highly innervated by axons from somatosensory and motor related cortical areas, while the medial one receives cortical axons from visual, auditory, associative, limbic (*Hintiryan et al., 2016*; *Hunnicutt et al., 2016*) and with lower axonal density from somatosensory cortical regions (*Reig and Silberberg, 2014*).

In addition to the differences in cortical or thalamic axonal innervation, there are dissimilarities in the composition of the striatal microcircuits. The distribution of parvalbumin (PV) and cholinergic interneurons (ChIs) along the mediolateral axis of the DS is not homogeneous (*Gerfen et al., 1985*; *Kita et al., 1990*; *Matamales et al., 2016*; *Muñoz-Manchado et al., 2018*). Furthermore, dopaminergic projections from the lateral part of the substantia nigra massively innervate the DS (*Ikemoto, 2007*), with particular impact in the activity of ChIs along the DS (*Chuhma et al., 2018*). All these precise afferent connectivity, microcircuit interactions, and neuromodulation regulate the synaptic activity of DLS- and DMS-MSNs.

The activity of every neural circuit is limited by anatomical and functional constrains which will restrict the repertoire of spontaneous and evoked activity patterns, defining the functional connectivity of the brain (*Getting, 1989*; *Luczak et al., 2015*). In this work, we describe how MSNs of the lateral and medial regions of the DS integrate spontaneous and sensory evoked activity.

The slow wave oscillation (SWO) is characterized by periods of high spontaneous activity (Up states) intermingled with silent periods (Down states) at the frequency of ~ 1 Hz, which is originated in the cortex (*Sanchez-Vives and McCormick, 2000*; *Timofeev et al., 2000*) and propagates directly to the striatum, modulating the resting state of MSNs (*Sáez et al., 2018*; *Wilson and Kawaguchi, 1996*) and interneurons (*Reig and Silberberg, 2014*). Based in their heterogeneous activity, we found that DS is segregated in two circuits and propose a biological substrate that explains their differences.

Because MSNs recorded in vivo are known to fire scarcely (*Adler et al., 2012*; *Berke et al., 2004*; *Wilson, 1993*), we performed single and pairs of whole-cell patch-clamp recordings to analyze their subthreshold dynamics during spontaneous and evoke activity, identifying their specific pathways. Our findings show how the DS is divided in two non-overlapping circuits, based on the MSNs activity. DLS- and DMS-MSNs differ in the integration of the slow wave and beta oscillations, as well as in the functional coupling to multiple cortical regions. By means of double in vivo patch-clamp recordings, we demonstrate a sequential propagation of the cortical slow wave oscillation (SWO) along DS. In addition, we found that the evoked responses of MSNs to visual stimulation displayed different properties along the medio-lateral axis, that were consistent with cortical projections and independent of the circuit in which the MSN was embedded. MSNs close to the midline responded with shorter delays, bigger amplitudes and faster slopes than the ones placed in dorsocentral territories. Finally, we identified that the direct and indirect pathways MSNs have particular attributes in the DLS and DMS, displaying differences in their electrophysiological properties and synaptic integration.

In conclusion, consistent with previous behavioral studies, our results demonstrate that DS is divided in two functional circuits, separated by a sharp boundary, each of them with specific properties that are essential to understand the striatum and basal ganglia functions.

## Results

### MSNs in the DLS and DMS have different electrophysiological properties

We obtained in vivo whole-cell patch-clamp recordings from 223 neurons located in the DS (n = 197) and several cortical areas (n = 26). All of them displayed SWO with prominent Up and Down states (*Figures 1C*, *2A*, Figures 4A–B, D, 5A–B and 6A), at close frequency of ~ 0.7 Hz in both brain regions. The different types of striatal neurons were identified by their electrophysiological properties and morphology (*Figure 1B*, see Materials and methods). Direct and indirect pathway MSNs were determined by their responses to the light stimulation using the optopatcher (*Katz et al., 2019*; *Ketzef et al., 2017*; Figure 7A). All average graphs showed in this study represent the standard deviation unless stated otherwise.

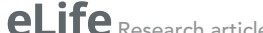

**Figure 1.** Analysis and classification of SWO in dorsal striatum. (**A**) Schematic representation of the in vivo recording setup. (**B**) Morphological reconstruction of DLS-MSN (left) and DMS-MSN (right). Different scales show neuron magnitude and its dendritic spines, confirming that the recorded neuron is a MSN. (**C**) Representative LFP (top) and whole-cell patch-clamp recording of a MSN (bottom). (**D**) Featurization of the SWO of DLS- and DMS-

*Figure 1 continued on next page*

*Figure 1 continued*

MSNs (see Materials and methods). (E) RFE of the computed features (feature number) to determine their order of relevance classifying DLS- and DMS-MSNs. A SVM with a linear kernel was selected as classification algorithm. (F) Example ROC curve of the DLS-DMS classification for one of the crossvalidations using the three most relevant features, illustrated in D. (G) Schematic representation of the data distribution of DLS and DMS groups for the most relevant features (numbers 11, 5, and 9, respectively). White number represent the mean; the radius of the circle represents the variance. (H) Subspace of classification based on the RFE results. Dots represent individual MSNs recorded in DLS (red) or DMS (blue). Classification hyperplane obtained after training the SVM with linear kernel in black. p Values obtained using the Wilcoxon Rank-sum test.

The online version of this article includes the following figure supplement(s) for figure 1:

**Figure supplement 1.** Classification of DLS- and DMS-MSNs at animal level.

**Figure supplement 2.** Correlation between the SWO features and electrophysiological properties of MSNs.

In order to study the circuit attributes that may characterize the DS, our initial approach was to compare the electrophysiological properties of striatal MSNs recorded in the lateral (DLS, n = 77) and medial (DMS, n = 91) regions of the DS in anesthetized mice. The membrane potential of the Down states was stable during the recordings with no differences between striatal regions (DLS-MSNs = −69.89 ± 5.79 Hz, DMS-MSNs = -69.867 ± 7.67 Hz, p=0.42). MSNs exhibited a rich sub-threshold activity during Up states; however, they displayed a very low rate of spontaneous action potentials with similar frequencies along DS (DLS-MSNs = 0.3 ± 0.95 Hz, DMS-MSNs = 0.24 ± 0.53 Hz, p=0.58). Input resistance was higher for DLS-MSNs (DLS-MSNs = 312 ± 25 MΩ, DMS-MSNs = 255 ± 16 MΩ, p=0.04). These changes in resistance could impact in the neuronal properties of synaptic integration, setting the gain and timing for their synaptic inputs. Thus, the integration of the spontaneous activity on DLS- and DMS-MSNs could be, at least partially, modulated by the cellular differences which underlie their intrinsic electrophysiological properties.

## The spontaneous activity from DLS- and DMS-MSNs have different attributes

We first studied the properties of the SWO recorded in whole-cell from dorsal striatal MSNs. In order to obtain a quantitative description of the activity of the MSNs in this brain state, we computed 13 different features to characterize the magnitude (i.e mean and max membrane potential, amplitude) and shape (i.e. ratio between transitions, number of peaks) of the Up states (*Figure 1D*), from which 11 were statistically different between DLS- and DMS-MSNs (Ups extraction and all features are explained in methods section). The average membrane potential in the Up states was higher in the DMS-MSNs, as well as their standard deviation, minimum and maximum membrane potential and the total amplitude (*Figure 1D*, features n° 1–5). Their upward and downward transition slopes and its ratio were also higher in DMS-MSNs (*Figure 1D*, features n° 8–10). DLS-MSNs displayed higher number of peaks during the Up states and peak to peak amplitude, as well as longer Up states (*Figure 1D*, features n° 11–13). Similar to cortical neurons (*Sanchez-Vives and McCormick, 2000*), Up states in MSNs are comprised by a barrage of excitatory and inhibitory synaptic inputs (*Reig and Silberberg, 2014*). Therefore, these divergences in the integration of the spontaneous activity may reflect changes in the excitatory and inhibitory inputs that DLS- and DMS-MSNs receive.

Once we obtained a statistical prove of the differences in the SWO between DLS and DMS, we asked whether these differences were sufficient to define DLS- and DMS-MSNs as two different populations. More specifically, we asked which parameters, or combination of them, could be used to distinguish between DLS- and DMS SWO more accurately. To do so, we searched for those features of the Up states whose combination maximized the accuracy of classifying MSNs corresponding to DLS or DMS circuits. Using a supervised machine-learning technique, named Support Vector Machine (SVM) (*Cortes and Vapnik, 1995*) with a linear kernel, we performed a Recursive Feature Elimination (RFE) analysis to rank the 13 computed features of the Up states, according to their combined utility to classify DLS- or DMS-MSNs (*Figure 1E*). Therefore, starting from the whole set of parameters, they were recursively removed depending on their relevance for the classification of DLS- and DMS-MSNs (see Materials and methods); as a results of the RFE, we obtained a sorting of the features depending on their relevance for the classification. The most relevant features were: first, the number of peaks in the Up states and second, the amplitude inside the Up state, which

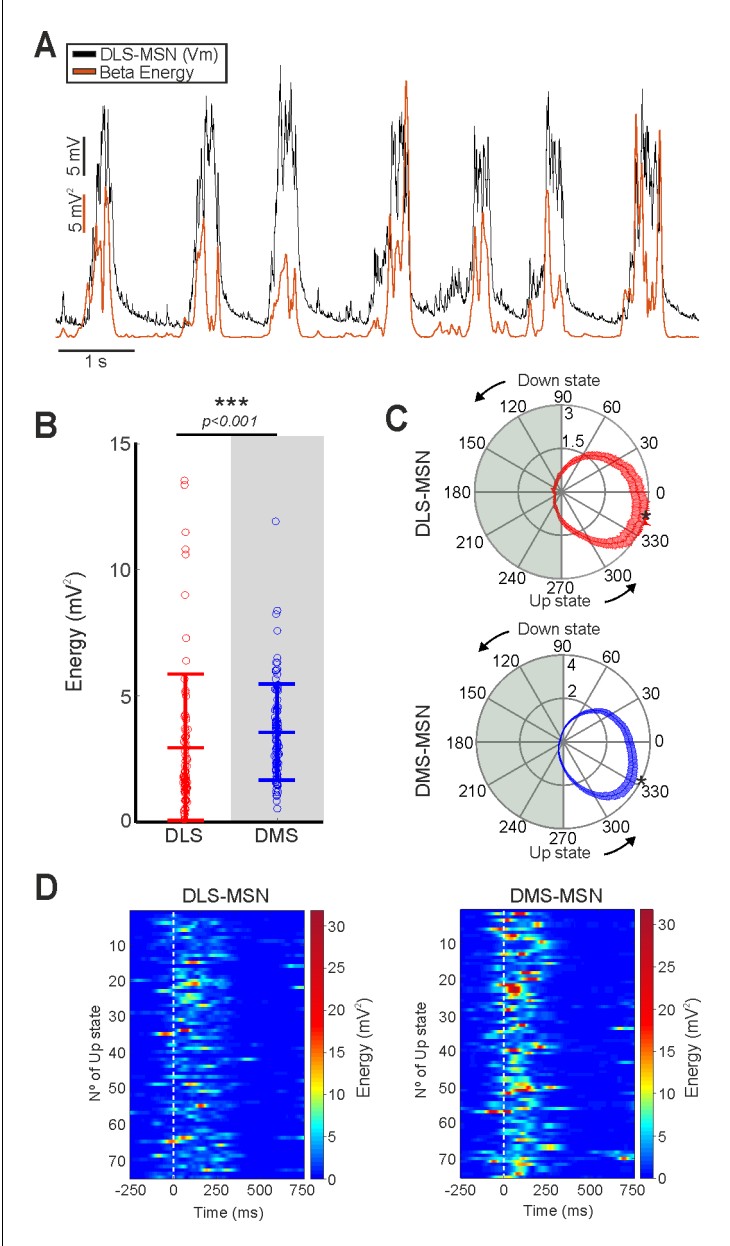

**Figure 2.** Beta band in membrane voltage of dorsal striatal MSNs. (**A**) Example of energy of the beta band (red trace) during the MSN SWO recorded in DMS. (**B**) Energy of the beta band of DLS- and DMS-MSNs. (**C**) Average of phase alignment of beta band to the SWO in DLS- (left) and DMS-MSNs (right) (Raylegh test, p<0.001 in both cases). Asterisk indicates the radial position of the beta peak. Beta peak occurs first in DMS-MSNs (p<0.001). (**D**) Representative examples of beta phase locking in DLS- (left) and DMS- MSNs (right). White line represents the beginning of each Up state. p Values in **B** obtained using the Wilcoxon Rank-sum test. p Values of the phase locking in **C** were computed using Rayleigh test.

The online version of this article includes the following figure supplement(s) for figure 2:

**Figure supplement 1.** Theta and Gamma band in membrane potential of dorsal striatum.

**Figure supplement 2.** Example of decomposition of different traces by the NA-MEMD.

resulted in an accuracy score of 84.61 ± 0.47%. When a third feature, the speed of transition from Up to Down states, was added, the accuracy increased to 88.56 ± 0.57%. The addition of new parameters increased the accuracy up to ~ 92%, value at which it remained stable (*Figure 1E*). In a compromise between accuracy and interpretability, we selected the space formed by the three most

relevant features for further analyses; number of peaks, amplitude and Up to Down transition slopes (*Figure 1H*).

Because multiple MSNs were recorded from the same animals (on average ~ 4 MSNs per mouse, *Supplementary file 1B*), and this could weigh the distributions and bias the result, we did an additional one-animal-out classification (*Figure 1—figure supplement 1*). In this analysis, all the MSNs recorded from one animal are excluded from the training phase of the classifier. Once trained, these MSNs are presented to the classifier in order to predict their position (see Materials and methods). By using this approach, we ensure that any bias that the animal may have introduced into the training phase of the classifier is removed. The result shows that the classification is present at the animal level (*Figure 1—figure supplement 1*), as well as the neuronal one (*Figure 1H*). In conclusion, our classifier based on the spontaneous activity integration of the MSNs indicates that DLS and DMS are identified as two distinct functional circuits.

In order to unravel the source of the Up states differences between DLS- and DMS-MSNs, we first computed the correlations of the intrinsic membrane properties, showed in *Table 1* and the Up state features of the MSNs; no comparison had an absolute correlation value over 0.3 (*Figure 1—figure supplement 2*). Therefore, it cannot be concluded that the intrinsic electrophysiological properties of MSNs are relevant to discriminate between striatal regions.

In the next step to further depict the differences between DLS and DMS, we characterized the fast oscillations that are phase locked to the Up state. High-frequency oscillations are prominent during wakefulness and during the Up states of the SWO (*Compte et al., 2008*; *Steriade et al., 1996*). In the cortico-basal ganglia-thalamic loop, high-frequency activity has been related with different functions and processes in health and disease (*Brown, 2003*; *Feingold et al., 2015*). For instance, exaggerated beta-band oscillations occur in Parkinson's disease. With the purpose to study possible differences in the high-frequency oscillations of DLS and DMS, we compared the energy in the theta (6–10 Hz), beta (10–20 Hz) and gamma (20–80 Hz) bands of DLS- and DMS-MSNs (*Figure 2—figure supplement 1*). To that end, we applied the Noise-Assisted Multivariate Empirical Mode Decomposition (NA-MEMD), extracting the Intrinsic Mode Functions (IMFs) (see Materials and methods) carrying the information of the different oscillatory activity bands. The NA-MEMD is a noise-assisted template-free technique for time frequency analysis of n-dimensional signals; it let us obtain an enriched decomposition of the spontaneous activity compared to traditional techniques (Fourier decomposition, wavelet, and others), due to the non-linear properties of neural oscillations (*Averbeck et al., 2006*; *Cole and Voytek, 2017*; *Laurent, 1996*; *Shamir and Sompolinsky, 2004*).

Our results show that the energy of the beta band was higher, and statistically different, in DMS-MSNs (*Figure 2B–D*). On average, the peak of beta energy occurs in the first third of the Up state in both striatal regions, but it is closer to the beginning of the Up state for DMS-MSNs (*Figure 2C*, DMS=-26.81±24.63°, DLS=-13.26±30.01°, p=0.0007). Following this divergence in beta band, we tested whether high-frequency oscillation contained enough information to classify DLS- and DMS-

**Table 1.** Intrinsic properties of DLS and DMS of direct and indirect MSNs.
Comparisons between DLS- and DMS-MSNs (p<0.05, *symbol). Comparisons between indirect DLS-MSNs and indirect DMS-MSNs (p<0.01, # symbol). All values are means ± SEM. p Values obtained using the Wilcoxon Rank-sum test.

| | Input Resistance (MΩ) | Resistance Down state hyp. (MΩ) | Resistance Down state dep. (MΩ) | Resistance Up state hyp. (MΩ) | Resistance Up state dep. (MΩ) | Capacitance (pF) | Tau (ms) |
|---|---|---|---|---|---|---|---|
| DLS | 312± 25 * | 292± 24 * | 313± 23 * | 304± 23 | 315± 27 | 19.45± 1.52 | 4.57± 0.23 |
| dMSNs | 297± 33 | 279± 31 | 284± 28 | 289± 30 | 296± 35 | 20.72± 1.98 | 4.52± 0.29 |
| iMSNs | 338± 41 # | 312± 36 # | 355± 40 # | 324± 37 | 342± 41 | 17.58± 2.35 | 4.63± 0.40 |
| DMS | 255± 16 * | 241± 20 * | 259± 17 * | 262± 20 | 262± 17 | 21.68± 1.55 | 4.61± 0.15 |
| dMSNs | 260± 22 | 249± 28 | 265± 23 | 269± 28 | 267± 23 | 23.20± 2.21 | 4.77± 0.19 |
| iMSNs | 241± 18 # | 223± 18 # | 248± 16 # | 248± 20 | 252± 20 | 18.43± 1.04 | 4.26± 0.24 |

MSNs. To that end, we trained an SVM with a linear kernel with the amplitude and position of the peak energy inside the Up state of each band. We obtained a classification accuracy of 70.61 ± 0.89% indicating that the high frequencies do not discriminate between DS regions as clearly as Up state properties. Moreover, the addition of the high-frequency properties as extra features to our previous classifier did not improve the level of discrimination obtained by the SWO features (*Figure 1E,F*). Therefore, we conclude that DLS- and DMS-MSNs can be identified as belonging two different circuits based on the features of the Up states during the SWO.

## DLS and DMS are two non-overlapping functional circuits in mouse

Once it had been shown that DLS and DMS circuits could be separated based on their SWO, we studied the transition from one circuit to the other, exploring the properties of this brain state in dorsocentral striatum (DCS) (*Figure 3A*), a hypothetical third region between DLS and DMS (see Materials and methods). We considered three possible scenarios (*Figure 3A,B*): If the activity of both circuits is mostly driven by the afferent inputs, we should observe a gradient moving from one circuit to the other (Hypothesis 1), or alternatively a third type of circuit (Hypothesis 2). Whether other factors, either extrinsic or intrinsic to the DS, control the spontaneous activity of these circuits, we should observe a sharp transition from one circuit to the other (Hypothesis 3). To address this question, we recorded MSNs in the DCS (see Materials and methods) in whole-cell, extracting the previous 13 features of the SWO (*Supplementary file 1A*). Then, we projected the DCS-MSNs into the standardized space used for the classification of DLS- and DMS-MSNs (*Figure 3C*), containing the values of number of peaks in the Up state, Up state amplitude and Up to Down transition slopes (*Supplementary file 1A*, *Figure 3—figure supplement 1*). In order to study the distribution of DCS-MSN in the parameter space relative to the populations of DLS- and DMS-MSNs we used the SVM that had been used previously to classify these two populations. We wanted to understand if the DCS-MSNs created a continuum between the DLS- and DMS-MSNs populations along the decision axis or whether this separation remained after the addition of the new pool of MSNS. To answer this question, we used the hypervector computed by the SVM to perform the classification (*Figure 3C*, see Materials and methods). We projected the data onto this hypervector and compared the distribution of the MSNs recorded from DCS to determine whether they belonged to either DLS or DMS circuits, or whether they had different properties (*Figure 3D*). If a sharp functional boundary exists between DLS and DMS, we would have recorded an undetermined amount of MSNs belonging to both circuits (Hypothesis 3). Otherwise, it would appear a new cluster with an undescribed third functional type of MSN, with an unknown distribution along the hypervector (Hypothesis 2) or in between DLS and DMS-MSNs distributions (Hypothesis 1). The distribution of MSNs recorded in the DCS was consistent with a mixture of MSNs from DLS and DMS circuits supporting a sharp transition between DLS and DMS regions [see discussion]. DCS distribution (*Figure 3D*, green trace) was significantly different to either the DLS and DMS distributions (p<0.01, *Figure 3D*, red, blue, and black traces respectively) and equivalent to the combination of both (p=0.681, *Figure 3D*, green and black traces).

Thus, we conclude that all recorded MSNs in DS belong to the previously described DMS and DLS functional circuits and they are separated by a sharp functional boundary.

## Sensory response in the boundary between circuits

MSNs in the DS integrate bilateral and multisensory information (*Reig and Silberberg, 2014*). This process is supported by an anatomically restricted distribution, depending on the recipient area of axons coming from cortical sensory regions. Bilateral tactile responses to whisker deflections can be recorded along the DS, from the most lateral territories to the border of the lateral ventricle in the DMS. On the other hand, visual responses seem enclosed to medial territories (*Reig and Silberberg, 2014*). Inspired by this description, we aimed to discern whether MSNs located in the anatomical region where the boundary between DLS and DMS was observed responded to visual stimulation and how they were related with our previous DLS- or DMS-MSNs classification. To this end, visual stimuli were presented to the contralateral eye as 15 milliseconds flashes from a white LED (see methods) during whole-cell recordings in the 17 MSNs located in the DCS. Our result shows that eight of them (47%) responded to the visual stimulation (*Figure 3E*). Then, we asked how these neurons were labeled by our classifier (*Figure 3C*): From nine neurons identified as DMS-MSNs, five of



**Figure 3.** Study of the SWO in the DCS striatum and determination of the functional boundary between DLS and DMS. (**A**) Schematic representation of the 'boundary question'. (**B**) Three possible hypotheses about DCS-MSNs SWO. Hyp. 1: There is an intermediate distribution between DLS and DMS SWO (top); Hyp. 2: There is a different type of SWO distribution (middle); Hyp. 3: There is no specific DCS distribution of SWO and the MSNs recorded at DCS are a combination of DLS- and DMS-MSNs (bottom). The black line displays the distribution of the SWO combining both DLS and DMS. (**C**) Distribution of DLS-, DMS-, and DCS-MSNs in the subspace of classification determined by the RFE. Classification plane in black. Orthogonal hypervector to the classification plane in gray. (**D**) Distribution of dorsolateral, dorsomedial and dorsocentral MSNs and the combining DLS-DMS function along the orthogonal hypervector to the hyperplane classification. All comparison between distributions are significant (p<0.01) except for DCS with DMS-DLS (p=0.681). (**E**) Percentage of neurons recorded in the DCS coordinate, responding or not to visual stimulation and classified as DLS or DMS. (**F**) Waveform average of visual responses recorded in an MSN from DCS (green) and DMS (blue). (**G–I**) Averages of onset (**G**), amplitude (**H**), and slope (**I**) of the visual responses recorded in MSNs from DCS coordinate and DMS. p Values obtained using the Wilcoxon Rank-sum test.

The online version of this article includes the following figure supplement(s) for figure 3:

**Figure supplement 1.** Z-scoring of the SWO features.

them responded to the visual stimulus; and from eight neurons identified as DLS-MSNs, three of them responded as well (*Figure 3E*). Therefore, some neurons identified as DLS-MSNs responded to visual stimulation in the DCS coordinate.

Anatomically, it has been described that several visual areas project to the DS. For example, V1 projections are located in the most medial regions of the DS, while medial and lateral anterior visual cortical areas extend their projections to more lateral striatal territories (*Hintiryan et al., 2016*), including our DCS anatomical coordinate. This suggests that MSNs will respond to visual stimulation differently, depending on their position in the DS. Therefore, we recorded a new set of MSNs in the DMS responding to the same visual stimulation (n = 10) and compared them with respect to the MSNs with visual responses in the DCS coordinate (n = 8). The result (*Figure 3F–I*) shows that

amplitudes and slopes were statistically different and clearly higher in the DMS-MSNs (amplitudes DCS = 8.84 ± 3.15 mV, DMS = 18.67 ± 3.86 mV, p=0.00009; slopes DCS = 139 ± 48.1 mV/s, DMS = 432 ± 302.9 mV/s, p=0.0003). Importantly, the onset of the visual responses was nine milliseconds slower for the DCS-MSNs (DCS = 64.84 ± 5.49 ms; DMS = 55.02 ± 8.37 ms, p=0.007) (*Figure 3G*), suggesting that they receive inputs from different populations of neurons. This result is consistent with the difference in cortico-striatal projections from visual areas (*Hintiryan et al., 2016*), in which V1 axons project to the most medial region of the striatum, while they are absent in the DCS coordinate. On the other hand, axons from anterior medial (AM) and anterior lateral (AL) visual regions, related with other aspects of the visual processing (*Garrett et al., 2014*; *Marshel et al., 2012*) reach the central region of the DS with a smaller density of projections than V1 to DMS (*Hintiryan et al., 2016*). This is compatible with the smaller amplitudes and slower slopes (*Figure 3H,I*). Thus, we support that the differences in the visual response are due to the different visual regions that are projecting to those MSNs, independently of whether they belong to DLS or DMS circuitry.

## DLS- and DMS-MSNs segregation is explained by their membrane potential transitions

To understand the main source of differences between the integration of the spontaneous activity in the DLS- and DMS-MSNs, we started by analyzing the first ranked feature: number of peaks during Up states (*Figure 1D–E*, feature n°11). We studied whether this difference occurred in cortical regions, from where they can be transmitted to the DS (*Figure 4A*). It is known that striatal oscillations depend on the cortical ones (*Kasanetz et al., 2002*; *Wilson, 1993*) and in vitro slices containing the striatum are absent of spontaneous oscillations (*Planert et al., 2013*).

We performed in vivo whole-cell recordings in Layer V of FrA (frontal associative cortex), M1 (primary motor cortex), S1 (primary visual cortex), and V1 (primary visual cortex) (*Figure 4B*). We found that S1 and M1 have significantly higher average number of peaks per Up state than FrA and V1 (*Figure 4C*). This result together with previous anatomical descriptions (*Alloway et al., 2006*; *Hoffer et al., 2003*; *Hooks et al., 2018*) strongly suggest that the membrane dynamics inside the Up state in the DLS are controlled by the sensory-motor cortical areas.

The main features for the classification of DLS- and DMS-MSNs (*Figure 1G*) were relative to state transitions (Up to Down transition) or intra-state transitions (number of peaks). In order to understand whether these differences may be caused by changes in the depolarizing and hyperpolarizing dynamics of MSNs, we studied the temporal properties of the membrane potential transitions. To analyze the convergence of synchronized hyper/depolarizing events onto MSNs that created detectable changes in the whole-cell recording, we first detected the sharp transitions of the voltage trace and phase locked them to the SWO cycle (*Figure 4D* [see Materials and methods]). Thus, we obtained a measure of the temporal dynamics of hyper/depolarizing events of the MSNs. Then, we computed their ratio for the DLS- and DMS-MSNs. We found differences in the depolarizing/hyperpolarizing ratio (DH ratio) of DLS- and DMS-MSNs aligned to the transitions between Up and Down states (*Figure 4E*). The DH ratio was significantly higher, implying that the total balance of inputs, onto DLS-MSNs was biased toward a bigger depolarization during the transition from the Down to the Up state and toward a larger hyperpolarization values during the transition from the Up to the Down state. This is consistent with the significant differences found for both Up states slopes (*Figures 1D* and *4F*, features 8–9).

In summary, our results show that the main differences between DLS- and DMS-MSNs spontaneous activity, selected by our classifier, are consistent with the number and the temporal distribution of the voltage fluctuations of the membrane potential during the Up states, suggesting that they are modulated by excitatory inputs from specific cortical areas and the striatal microcircuits interactions, mostly inhibitory.

## Sequential propagation of the cortical Up states to the striatum

The SWO has a cortical origin, from where it propagates to the striatum activating MSNs (*Kasanetz et al., 2002*; *Ketzef et al., 2017*; *Reig and Silberberg, 2016*; *Sáez et al., 2018*; *Wilson and Kawaguchi, 1996*) and interneurons (*Reig and Silberberg, 2014*). In order to clarify how cortical activity is related with the DS, we studied the correlations of the Up states between



**Figure 4.** Number of peaks and depolarized/hyperpolarized ratio of the SWO in dorsal striatum. (**A**) Detection of peaks (black arrows) in the Up state in a DLS-MSN (red) and a DLS-MSN (blue) (see Materials and methods). (**B**) Representative examples of SWO in different cortical regions recorded. (**C**)

*Figure 4 continued on next page*

*Figure 4 continued*

Number of peaks per Up state in the different cortical regions (left). Non-labeled comparisons are not significant. (D) Extraction of depolarized/hyperpolarized events by computational approach (see Materials and methods). Top: Detection of the events using a threshold in the first derivative of the Vm. Scale bar represents 0.1 dmV/dt. Bottom: Representation of the detected depolarized (green) and hyperpolarized (orange) events from whole-cell recording of a DLS-MSN (middle). (E) Average of the depolarized/hyperpolarized ratio aligned to the SWO cycle of DMS- and DLS-MSNs. Comparison of positive (p<0.0072) and negative (p<0.0398) values at peaks. (F) Grand average of the transitions from Down to Up (upper) and Up to Down (bottom). Shaded bars in E and F represent SEM. p Values obtained using the Wilcoxon Rank-sum test. In C, alpha values for multiple comparisons were corrected using Holm-Bonferroni correction.

several cortical areas and MSNs located in DLS and DMS (*Figure 5*). First, we applied the NA-MEMD algorithm to extract the IMF carrying the SWO (*Figure 5A–B*). Then, we used it to compute the cross-correlation between the whole-cell recordings of the MSNs in DLS or DMS with the pairs of simultaneous local field potentials (LFPs) recorded in FrA, M1, S1, and V1. We found that neuronal activity from DLS- and DMS-MSNs had different correlations with FrA, M1 and V1 LFPs (*Figure 5D*). Correlation with FrA and M1 activity was higher for DLS neurons, while correlation with V1 was higher for neurons recorded from DMS. When we represented correlation values with FrA and V1, two clusters were clearly distinguishable, corresponding to DMS- and DLS-MSNs (*Figure 5E*), suggesting specific functional coupling to the cortex for both striatal regions. It is important to note that we used this approach to understand the global correlation of the striatal SWO with different cortical areas, which does not have to be confused with the modulation of activity once the MSNs are in the active Up state (*Figure 1C*, see Discussion).

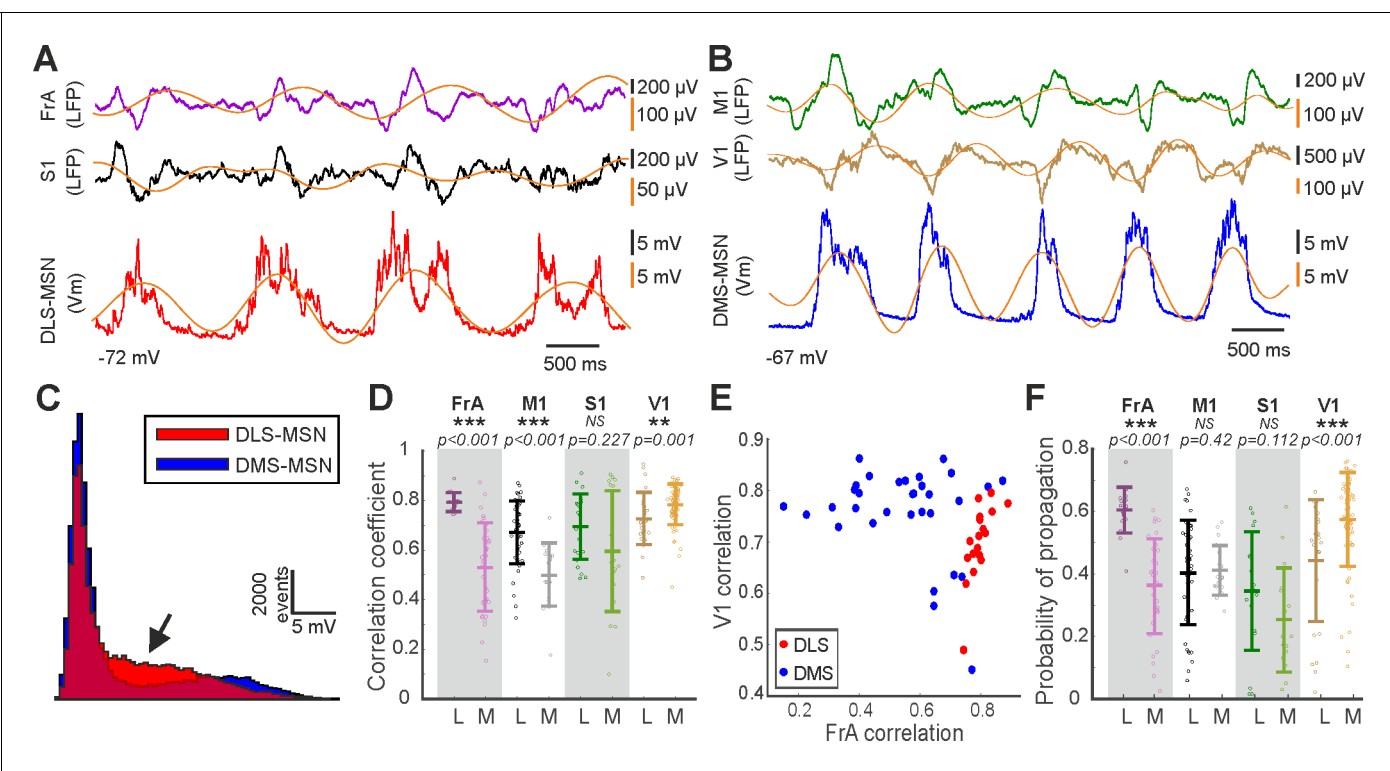

**Figure 5.** Integration of the cortical SWO in dorsal striatum. (A) Example of simultaneous recordings from a DLS-MSN (red) and double LFPs from FrA (purple) and S1 (black) together with SWO extraction by NA-MEMD (orange line). (B) Example of simultaneous recordings from a DMS-MSN (blue) and double LFP from M1 (green) and V1 (ochre) together with SWO extraction by NA-MEMD (orange line). (C) Histogram of the membrane potential values of the MSNs in A and B, the black arrow indicates the dissimilar shape of the bimodal distributions. (D) Correlation coefficient of different cortical regions to DLS- and DMS-MSNs. (E) Raster plot showing the correlation between all MSNs and the LFP from FrA and V1. (F) Probability of propagation of the Up state from different cortical regions to DLS- and DMS-MSNs. p Values obtained using the Wilcoxon Rank-sum test. In D and F, alpha values for multiple comparisons were corrected using Holm-Bonferroni correction.

Next, we used the IMF carrying the slow component of the SWO (*Figure 5A–B*, orange line) to analyze the probability of occurrence of an Up state recorded in an MSN, following the occurrence of a cortical Up state (*Figure 5F*). Consistent with the correlation results, the probability that an Up state in FrA is followed by an Up state in DLS was significantly higher than in DMS. Similarly, Up states in V1 had higher probability to be followed by an Up state in DMS than in DLS. In summary, the activity from FrA and sensory-motor cortical regions (M1 and S1) is strongly related with DLS-MSNs, while the activity in V1 does with DMS-MSNs.

These results predicted that, following the known propagation of the SWO across the cortex, with a predominant rostro-caudal direction (*Massimini et al., 2004*; *Ruiz-Mejias et al., 2011*), a sequential activation could be expected along the DS. In order to test this hypothesis, we first compared the order of transitions to an Up state recorded in both poles of the cortex by pairs of LFPs. We found an Up state in FrA preceding an Up state in V1 in the 71% of times (p value<0.0001, *Figure 6F*), demonstrating the preference of the rostro-caudal propagation of the SWO in cortex.

Finally, in order to test our prediction, we performed two simultaneous whole-cell patch-clamp recordings in two identified MSNs, located in the DLS and DMS (*Figure 6*). A representative example is shown in *Figure 6A–D*. In all six recorded pairs, the Up state in DLS-MSN preceded the one in DMS-MSN in 71.11 ± 9.37% of the times (*Figure 6E*, p=0.0313), with an average delay of

**Figure 6.** Sequential activation of the dorsal striatum during the SWO. (**A**) Representative example of simultaneous double in vivo whole-cell recordings in DLS (top, red) and DMS (bottom, blue). (**B**) Inset of the shaded part in **A**, showing two aligned membrane potential traces to their Down states. Notice that DLS-MSN onset is preceding the DMS-MSN. (**C**) Raster plot of the same example of a paired recording in DLS (top) and DMS (bottom) MSNs. Each line represents a 1 s time window aligned to the onset of each of the Up states of the DLS-MSN (white line). (**D**) Example of the distribution of delays between the onset of the Up state in the DMS- relative to the DLS-MSN. Positive values indicate that the Up state arrive later to the DMS-MSN. Same neuron in A, B, C, and D. (**E**) Directional probability of DS-MSNs, obtained from pairs of whole-cell recordings (N = 6). (**F**) Directional probability of V1 and FrA cortical areas, obtained from pairs of LFPs (N = 54). p Values obtained using the Wilcoxon Rank-sum test.

53.24 ± 32.16 ms. In the rest of the cases (28.62 ± 8.77%), in which DMS-MSN preceded the DLS-MSN, the delay was similar (average delay DLS-DMS = 51.91 ± 29.80 ms).

In conclusion, our results regarding the propagation of the cortical Up states to DS, confirm a stronger functional coupling of FrA to DLS and V1 to DMS. This generates a sequential activation of the DS, in which in most of the cases, DLS precedes DMS by tens of milliseconds.

## Direct and indirect MSNs have distinct properties in DLS and DMS

In a final step to deconstruct DS circuitry, we studied the activity of MSNs corresponding to the direct or indirect pathway. To understand corticostriatal dynamics it is essential to apprehend how both pathways integrate the upstream cortical activity. To answer our question, we analyzed the whole-cell recoding of the DS that were optogenetically identified as direct and indirect MSNs through the optopatcher (*Figure 7A*, see Materials and methods). First, we analyzed whether the features that were previously used to describe the Up states (*Figure 1D*) allowed us to differentiate between the MSNs belonging to each pathway in DLS and DMS. Our results show differences in the integration of the spontaneous activity between direct and indirect pathways only in DLS. We found three features of the Up states that were significant different when comparing direct and indirect MSNs in DLS: maximum and minimum membrane potential value of the Up state and the average

**Figure 7.** Differences between direct and indirect pathway MSNs in dorsal striatum. (**A**) Example showing an in vivo identification of an MSN using the optopatcher. Responses in D2-ChR2-YFP mice (top trace, ChR2$^+$, green) to light pulses, inducing depolarization in the MSN. Negative cells (bottom trace, ChR2$^-$, black) did not respond to light pulses. Blue squares indicate the intensity of the light pulse stimulation, from 20 to 100%. (**B**) RFE to optimize the classification of dMSNs and iMSNs in DLS (red) or DMS (blue). A SVM with a linear kernel was selected as classification algorithm. (**C**) Significant differences in three SWO features were found in DLS (red) between direct (dark) and indirect (light) pathways, but not in DMS (blue). p Values obtained using the Wilcoxon Rank-sum test.

The online version of this article includes the following figure supplement(s) for figure 7:

**Figure supplement 1.** Beta, Theta, and Gamma bands in membrane voltage of direct and indirect pathways in the DLS and DMS.

value of the Up state (*Figure 7C*). Then, we asked whether DLS- and DMS-MSNs could be subdivided in two additional populations corresponding with the direct or indirect MSNs in both striatal regions. The result did not yield separation, classification accuracy was 50% for DMS-MSNs and 64% for DLS-MSNs (*Figure 7B*). High-frequency oscillations were also similar between pathways in both striatal regions (*Figure 7—figure supplement 1*).

Finally, we studied whether the differences in the electrophysiological properties between DLS- and DMS-MSNs, described before (*Table 1*), were dependent on one of the two pathways. Resistance values, during Down states, were significantly higher for indirect DLS-MSNs compared to indirect DMS-MSNs, while no differences were found between direct DLS- and DMS-MSNs (*Table 1*).

In summary, our results show that DLS- and DMS-MSNs cannot be subdivided by the direct and indirect pathways. However, their differences in resistance and integration of the spontaneous activity demonstrate that direct and indirect MSNs have distinct properties in the DLS and DMS.

## Discussion

Circuits are limited but not defined by their anatomy; different axonal projections, cell types and synaptic dynamics delimit their spontaneous and evoked activity patterns. In this study, we have described two functional circuits in the anatomically homogenous DS, covering the medial and lateral regions and separated by a sharp functional boundary. We discriminated DLS and DMS circuits based on different properties of their spontaneous activity recorded in MSNs by means of the in vivo whole-cell patch-clamp technique. The use of an anesthetized preparation was convenient for a number of reasons; first, it granted the mechanical stability that allowed the recording of our dataset that includes whole-cell patch clamp recordings in multiple cortical and striatal areas, specially the double patch-clamp recordings in the striatum. In addition, during anesthesia, as well as sleep and resting awake periods, the brain activity is in the SWO regime (*Poulet and Crochet, 2018*). This stereotyped brain state is highly characterized extra- and intracellularly (*Sanchez-Vives et al., 2017*), and consists on slow traveling waves of cortical origin that will propagate to multiple cortical and subcortical areas, including the striatum. This approach allows the study of the functional constrains in the striatum that will modulate this spontaneous activity and are still present during awake or evoked states (*Getting, 1989*; *Luczak et al., 2015*), when striatal activity is more desynchronized (*Mahon et al., 2006*). Thus, the SWO is an adequate brain state to understand the functional organization of the DS.

To analyze the oscillatory activity, we used NA-MEMD algorithm (*Rehman and Mandic, 2010*) together with Hilbert transform (*Huang et al., 1998*). This algorithm is suited to decompose nonlinear nonstationary signals (*Alegre-Cortés et al., 2017*; *Alegre-Cortés et al., 2016*; *Hu and Liang, 2014*; *Mandic et al., 2013*). Given the well-known nonlinear properties of neural oscillations (*Averbeck et al., 2006*; *Cole et al., 2017*; *Laurent, 1996*; *Shamir and Sompolinsky, 2004*), the use of NA-MEMD leads to an increased detail of description when compared with traditional techniques (*Alegre-Cortés et al., 2017*; *Hu and Liang, 2014*). In our knowledge is the first time that it is used to analyze membrane oscillations of single neurons.

In humans, the dynamics of the SWO have demonstrated to be highly reproducible within and across subjects, providing information about the general state of the cerebral cortex and suggesting its applicability as a method to unravel functional circuits (*Massimini et al., 2004*). The analysis of the SWO has been used to detect neurophysiological alterations in humans and in a mouse models of neurological disorders such as; Alzheimer's disease (*Busche et al., 2015*), Down syndrome (*Ruiz-Mejias et al., 2016*), or epilepsy (*Amiri et al., 2019*), among others. Here, we analyzed the electrophysiological properties of MSNs, their evoked visual responses and the slow and the fast components of their oscillatory activity, and we found that the Up states of the SWO are useful method to identify brain circuits. Therefore, we are extending the previous concept and proposing a methodology to study the brain functional connectivity.

### DLS and DMS are two different functional circuits in mouse

We have shown that DS is divided in two different circuits, demonstrating the presence of a sharp functional transition from one circuit to another in the central region of the DS. This sharp transition is unexpected given the anatomical organization of corticostriatal projections, due to the profusion of axon terminals which provide overlapping inputs from multiple cortical areas to several regions of

the DS (*Alloway et al., 2006*; *Hintiryan et al., 2016*; *Hoffer et al., 2003*; *Hoover et al., 2003*). Hence, an essential question arises: why do we find a sharp boundary instead of a gradient transition between DLS- and DMS-MSNs? Our results show that the number of peaks during Up states is a critical feature to discriminate between DLS- and DMS-MSNs (*Figure 1E*), being higher in DLS-MSNs (*Figure 1D*, feature 11). In order to understand this result, we recorded layer V neurons in FrA, M1, S1, and V1, showing a larger number of peaks during Up states in FrA, M1, and S1 than in V1 (*Figure 4C*). This suggests that, in agreement with previous anatomical studies (*Alloway et al., 2006*; *Hoffer and Alloway, 2001*; *Hooks et al., 2018*), somatosensory-motor and premotor cortical regions exert sharper modulation onto the DLS-MSNs than onto DMS-MSNs, which is more anatomically connected with visual, associative, and limbic prefrontal cortical areas (*Hintiryan et al., 2016*; *Hunnicutt et al., 2016*). We also observed a higher number of peaks between M1-S1 with respect to FrA (*Figure 4C*). These results suggest that the dynamics of the excitatory/inhibitory inputs during the Up states change between cortical areas and subsequently influence the striatum, opening an interesting question regarding the circuit mechanism underlying the observed cortical Up states differences.

Together with the cortical glutamatergic inputs, several additional factors may contribute to generate a functional boundary. Our classification reached ∼ 90% of accuracy between DLS- and DMS-MSNs when the Up to Down state transition slopes were added (*Figure 1E*). Moreover, the Down to Up transition and transition ratio were both significantly different between DLS- and DMS-MSNs as well (*Figure 1D*). Therefore, Up states slopes are key elements to discriminate between dorsal striatal circuits too. In order to better understand the undergoing dynamics that could explain the differences in the slopes, we measured the time course of depolarized and hyperpolarized events of the MSNs during the SWO (*Figure 4D*) and compared them using the DH ratio (*Figure 4E*). Our results show more positive and negative ratio to the DMS-MSNs during the upward and downward Up states transitions slopes respectively, suggesting a different EI balance in the DLS compared to DMS. Cortical SWO propagates to both types of MSNs but also to FS and ChI (*Reig and Silberberg, 2014*). FS interneurons provide strong feedforward inhibition to MSNs (*Gittis et al., 2010*; *Szydlowski et al., 2013*; *Tepper et al., 2008*). Previous reports demonstrated the presence of a FS gradient (*Gerfen et al., 1985*; *Kita et al., 1990*; *Monteiro et al., 2018*) or diverse types of PV interneurons with different electrophysiological properties along DS (*Monteiro et al., 2018*; *Muñoz-Manchado et al., 2018*). In addition, in cortical slices with spontaneous SWO both upward and downward transition slopes are controlled by GABA$_A$ and GABA$_B$ receptor activation (*Perez-Zabalza et al., 2020*; *Sanchez-Vives et al., 2010*). Based on these evidences, and beyond the glutamatergic inputs, we hypothesize that PV interneurons may greatly influence the Up states slopes of DLS-MSNs making them different from DMS ones.

In addition, other mechanisms can also contribute to the generation of a functional boundary; for instance, a similar scenario exists for ChIs, which present a functional gradient with higher activity in the medial regions of the striatum (*Abudukeyoumu et al., 2019*; *Matamales et al., 2016*). ChIs could modulate differently the SWO in DLS- and DMS-MSNs by disynaptic inhibition (*English et al., 2012*). Under anesthesia, dopaminergic neurons of the Substantia Nigra discharge spontaneous action potentials, in tonic or burst firing mode (*Aristieta et al., 2016*; *Brown et al., 2009*). This dopaminergic activity could induce different effects in the SWO in DLS- and DMS-MSNs, either directly mediated by distinct dopaminergic projections (*Brown et al., 2009*) or indirectly by their varied impact in ChIs activity along the mediolateral axis (*Chuhma et al., 2018*). Finally, it has been recently shown that SOM corticostriatal interneurons modulate the MSNs activity from motor cortical areas (*Melzer et al., 2017*; *Rock et al., 2016*). This type of interneuron could regulate the Up states transitions slopes directly, inhibiting the MSNs in their distal dendrites. Future work will have to determine whether FS, ChI, SOM or maybe other types of interneurons and neuromodulators contribute to the generation of the sharp functional boundary between DLS and DMS.

## DLS and DMS are not defined by their sensory responses

Previous studies demonstrated that MSNs in the DS respond to sensory stimulation (*Ketzef et al., 2017*; *Pidoux et al., 2011*; *Reig and Silberberg, 2014*; *Sippy et al., 2015*). MSNs in the lateral and medial regions of the DS are activated by whisker stimulation, with bigger and sharper responses in the lateral region, which receives greater density of axons from S1. On the other hand, visual responses seem restricted to more medial territories, consistent with their corticostriatal projections

(*Reig and Silberberg, 2014*). Because the detected functional border is located in between these two striatal regions, we explored whether the previously described differences in the response to visual stimulation in DMS and DLS (*Reig and Silberberg, 2014*) was also present in the MSNs recorded in our DCS striatal coordinate, and therefore could let us to discriminate between DLS- and DMS-MSNs along the whole medio-lateral axis. Tactile-whisker stimulation was discarded because MSNs from lateral and medial regions respond to this sensory modality, with no differences in onset delay (*Reig and Silberberg, 2014*). Therefore, we tested whether neurons recorded in the functional border between DLS and DMS and classified as DLS- or DMS-MSNs responded to visual stimulation. We found that the 47% of MSNs recorded in the DCS coordinate responded to visual stimulation and the 17% were also labeled as DLS-MSNs by our classifier (*Figure 3E*). Then, we compared visual responses between MSNs recorded in the DCS and DMS coordinates. We found that they differ in their responses, including their onset delay (*Figure 3F–I*): MSNs in the DMS coordinate responded 9 ms earlier than the ones in the DCS coordinate. This strongly suggests that cortical axons sending visual information to the striatum have diverse cortical origin. A previous anatomical study *Hintiryan et al., 2016* demonstrated that the striatal region in which our DCS coordinate was located receives axons from the anteromedial (AM) and anterolateral (AL) areas of the visual cortex, while the DMS one does from V1, AM, and AL. In the hierarchy of visual areas, V1 sends abundant feedforward projections to AM and AL (*Garrett et al., 2014*), which could underlie the observed differences in onset delays. Thus, bringing together our results and the previous anatomical description, we hypothesize that, while the most lateral territories of the DLS are not involved in visual processing, there is a disparity of axons from several visual areas sending different information along the DS, which does not overlap with the internal organization of the DS in DLS and DMS. The functional specialization of V1, AM, and AL cortical areas compromise different properties of the visual information (*Marshel et al., 2012*), most probably, transmitting their particular attributes to the DLS- and DMS-MSNs that they contact. Future works are required to fully understand the spectra of sensory responses along the mediolateral axis of the DS. For instance, it remains as an open question a detailed understanding of how the specific properties of somatosensory or visual stimuli, are encoded along the latero-medial axis of the striatum.

In conclusion, the functional circuitry of the striatum cannot be described based on their responses to sensory stimulation. The DS is divided in two regions as a result of the large diversity of corticostriatal and probably other afferent projections, together with differences at the level of their microcircuits. On the other hand, while sensory inputs may not determine the overall dynamics of DLS or DMS circuitry, they govern the sensory responses of the MSNs in the specific striatal coordinates on which they project.

## Propagation of the cortical Up states to the striatum

SWO propagates along the cortical network (*Sanchez-Vives and McCormick, 2000*) and is transmitted to the striatum (*Reig and Silberberg, 2014*; *Wilson and Kawaguchi, 1996*) as well as to other subcortical nuclei (*Ros et al., 2009*; *Steriade et al., 1993*). Decortication or disruption of the cortical SWO impairs the striatal ones (*Kasanetz et al., 2002*; *Wilson, 1993*) and on the other hand, striatal slices are absent of rhythmic spontaneous activity (*Planert et al., 2013*).

As occurs in human slow-wave sleep (*Massimini et al., 2004*), anesthetized mice exhibit a predominant pattern of SWO propagation from rostral to caudal brain poles (*Ruiz-Mejias et al., 2011*). Here, we show how the correlation and the probability of transition of an Up state from FrA is higher to the DLS, while an Up state recorded in V1 has higher probability to propagate to the DMS (*Figure 5D–F*). These results suggest a sequential activation of DMS and DLS, similar to the one between both poles of the cortex during the SWO. As discussed before, this result describes the corticostriatal propagation of the SWO, which is different of the modulation of MSNs activity during the Up states. In order to test this hypothesis, we simultaneously recorded pairs of MSNs in DLS and DMS, in our knowledge this is the first time that double in vivo patch-clamp recordings are shown in the striatum. We found a similar probability of an Up state to appear in DLS before DMS and to appear in FrA before V1 (*Figure 6*). This description of the overall dynamics of corticostriatal transmission is based on functional, rather than anatomical connectivity, supporting a stronger coupling of caudal, sensory related cortical regions to DMS, whereas premotor and sensory-motor frontal regions are more tightly connected to DLS. Consistent with this, the sequential activation of both DS regions during a two-forced-decision-task has been recently reported, in which DMS was first

activated during the sensory stimulation period, followed by an increase in DLS activity coinciding with decision making and motor preparation/execution (*Peters et al., 2019*).

Finally, the number of peaks during Up states, the first feature selected to classify between DLS- and DMS-MSNs, were prominent in the DLS-MSNs (*Figure 1D*) as well as in S1 and M1 (*Figure 4B–C*). These results together with the anatomical descriptions (*Alloway et al., 2006*; *Hoffer et al., 2003*; *Hooks et al., 2018*) and a recent in vivo synaptic transmission experiments (*Charpier et al., 2020*), strongly suggest that the activity inside the Up state of the DLS-MSNs is controlling by sensory-motor areas. In addition, FrA send copious projections to the DLS (*Hintiryan et al., 2016*; *Hunnicutt et al., 2016*) and we found that correlations and probability of propagation of Up states to DLS were higher for FrA than S1 and M1 (*Figure 5D–F*).

Considering this results together, we suggest that, when a slow wave starts in the rostral part of the cortex, FrA projections trigger the Up state in DLS, but rarely DMS. Then, once the wave reaches somatosensory-motor areas, they will modulate the activity of DLS-MSNs, which are already in an Up state. In agreement, it was shown that the blockage of synaptic transmission in S1 by TTX application during bilateral whisker stimulation blocked whisker responses but not the Up states in DLS-MSNs (*Reig and Silberberg, 2016*), demonstrating that their Up states were not triggered by S1. Therefore, we hypothesize that during awake states, FrA could act as a gain modulator of DLS that will facilitate its response to somatosensory-motor inputs.

## High-frequency oscillations

High-frequency oscillations are usually associated with wakefulness. However, they also occur during the SWO in natural sleep, under anesthesia (*Steriade et al., 1996*) and in isolated cortical slices (*Compte et al., 2008*). Especially relevant regarding the cortico-basal ganglia-thalamic loop are beta oscillations, which have been linked with the control of voluntary movements. Differences in Beta oscillations between caudate and putamen were reported in healthy monkeys at the end of a learned motor task (*Feingold et al., 2015*). Exaggerated Beta oscillations occur in Parkinson's Disease and in rats under dopamine depletion (*Brown, 2003*; *Mallet et al., 2008*; *Sharott et al., 2017*). Our results did not show differences between DLS- and DMS-MSNs for theta and gamma bands (*Figure 2—figure supplement 1*), however a difference in beta energy was detected (*Figure 2*). Possible candidates to explain the observed higher beta energy in DMS-MSNs are ChI. This type of interneurons are more active in the DMS (*Matamales et al., 2016*) and can facilitate glutamate release from presynaptic terminals to the MSNs (*Abudukeyoumu et al., 2019*). Thus, they might promote the fast glutamatergic transmission observed in DMS.

Our work adds further detail to the description of beta oscillation in healthy mice thanks to the use of NA-MEMD together with Hilbert transform (*Figure 2—figure supplement 2*), providing a new substrate for the study of aberrant beta oscillations in Parkinson disease. This result may indicate that the aberrant oscillations in Parkinson disease may be produced by the misbalance of normal striatal dynamics, as previously suggested (*McCarthy et al., 2011*).

We also analyzed the fast oscillatory activity to classify DLS- and DMS-MSNs, however this information was not useful for that purpose. Unlike other faster oscillatory bands, the ~ 1 Hz oscillation, which constitutes the main component of the SWO brain state, is described as highly reproducible within and across subjects. Based in this property, it was suggested as a method to study neuronal changes and connectivity in humans (*Massimini et al., 2004*). Our negative result using the fast oscillatory activity reinforces the idea that the ~ 1 Hz component of the SWO is a particular useful model to study functional connectivity.

## Integration of the spontaneous activity in the direct and indirect MSNs along the DS

Cortical inputs innervate the DS targeting both the direct and indirect pathway MSNs (*Doig et al., 2010*; *Kress et al., 2013*; *Wall et al., 2013*). Both types of MSNs are co-activated during action initiation (*Cui et al., 2013*) and respond to tactile and visual stimulation (*Reig and Silberberg, 2014*). However, a large number of studies described differences between direct and indirect MSNs in the DLS: dMSNs have higher density of cortical and thalamic afferent synapses (*Huerta-Ocampo et al., 2014*), bigger response to whisker stimulation (*Ketzef et al., 2017*; *Reig and Silberberg, 2014*; *Sippy et al., 2015*) and based on in vivo SWO, it has been estimated that dMSNs receive stronger

synaptic input than iMSNs (*Filipović et al., 2019*), among other differences. Here, we have described changes in the activity integration between pathways in the DLS-MSNs (*Figure 7C*), which are absent in the DMS-MSNs. Despite these differences, our results show that both DLS and DMS are unified circuits that involve both pathways.

Previous in vitro (*Planert et al., 2010*) and in vivo (*Ketzef et al., 2017*; *Reig and Silberberg, 2014*) studies in the DLS showed that iMSNs have higher input resistance than dMSNs. While observed a similar trend, the main difference in the input resistance was detected between DLS-MSNs and DMS-MSNs, with bigger values in DLS. Moreover, this result was weighted by the iMSNs located in the DLS, which displayed the highest values of resistance and were statistically different than iMSNs in the DMS (*Table 1*). Interestingly, this variation between iMSNs located in both striatal regions occurred during Down states, when MSNs are mainly silent, suggesting that changes in resistance are independent of the glutamatergic inputs. It is known that dopamine alters the input resistance of MSNs in vivo (*Ketzef et al., 2017*). Therefore, distinct dopaminergic innervation of the DLS- and DMS-MSNs (*Chuhma et al., 2018*) could underlie our differences in resistance between striatal regions.

## Conclusion

Human caudate and putamen have been compared to the rodent DLS and DMS, respectively (*Balleine and O'Doherty, 2010*), based in their corticostriatal connectivity and behavioral functions. Yet, a study of the circuitry that supported this division was necessary. Here, we have shown how DLS and DMS are two non-overlapping circuits isolated by a sharp functional boundary. This work provides further understanding of the corticostriatal organization and reveals the biological substrate to divide the DS in two different circuits. We have shown how DLS- and DMS-MSNs display independent spontaneous regimes during SWO brain state, that can only be explained by a combination of particular corticostriatal functional connectivity and microcircuit properties. Visual evoked responses in particular demonstrated that the differences between DLS and DMS circuits cannot be reduced to their interaction with sensory cortices. In conclusion, our results provide the required understanding to support that this functional segregation is analogous to the anatomical and functional division of the primate striatum in caudate and putamen. Considering the relevance of the mouse striatum as a model for multiple human diseases, our results indicate that research will have to consider the idiosyncrasy of the two regions of the DS.

# Materials and methods

**Key resources table**

| Reagent type (species) or resource | Designation | Source or reference | Identifiers | Additional information |
|---|---|---|---|---|
| Genetic reagent (*M. musculus*) | BAC-Cre Drd2-44 or STOCK Tg(Drd2-cre) ER44Gsat/Mmcd | GENSAT | RRID:MMRRC_ 017263-UCD | Males and females used |
| Genetic reagent (*M. musculus*) | Ai32 or Ai32(RCL-ChR2(H134R)/EYFP) or B6;129S-Gt(ROSA) 26Sortm32(CAG-COP4*H134R/EYFP)Hze/J | The Jackson Laboratory | Stock No: 012569 | Males and females used |
| Genetic reagent (*M. musculus*) | C57BL/6J or C57BL/6NCrl | Charles River Laboratories | Strain Code: 027 | Males and females used |
| Other | Cy3 conjugated streptavidin | Jackson ImmunoResearch Laboratories | Cat#: 016-160-08 Lot. #125000 | 1:1000 |
| Chemical compound, drug | Ketamine, Ketamidor | Alvet Escartí S.L. | Ref. # 078100377 | 100 mg/ml |

*Continued on next page*

*Continued*

| Reagent type (species) or resource | Designation | Source or reference | Identifiers | Additional information |
|---|---|---|---|---|
| Chemical compound, drug | Medetomidine, Sedine | Alvet Escartí S.L. | Ref. # 005100740 | 10 ml |
| Chemical compound, drug | Sodium Pentobarbital, Dolethal | Alvet Escartí S.L. | Ref. # 015P5502 | 200 mg/ml, 100 ml |
| Software, agorithm | Spike2 | Cambridge Electronic Design Limited (CED) | n/a | Version 9 |
| Software, algorithm | Matlab | Mathworks | n/a | Version 2018 |
| Software, algorithm | Support Vector Machine (SVM) | *Cortes and Vapnik, 1995* doi: https://doi.org/10.1007/BF00994018 | n/a | https://www.scipy.org/ |
| Software, algorithm | NA-MEMD | *Rehman and Mandic, 2010* doi: https://doi.org/10.1098/rspa.2009.0502 | n/a | http://www.commsp.ee.ic.ac.uk/~mandic/research/emd.htm |

## Ethical permit

All the experimental procedures were conformed to the directive 2010/63/EU of the European Parliament and the RD 53/2013 Spanish regulation on the protection of animals use for scientific purposes, approved by the government of the Autonomous Community of Valencia, under the supervision of the *Consejo Superior de Investigaciones Científicas* and the Miguel Hernandez University Committee for Animal use in Laboratory.

## Animal model

D2-Cre (ER44 line, GENSAT) mouse line was crossed with the Channelrhodopsin (ChR2)-YFP reporter mouse line (Ai32, the Jackson laboratory) to induce expression of ChR2 in indirect MSNs and was used to perform optogenetic differentiation of direct (dMSN) and indirect (iMSNs) MSNs (*Ketzef et al., 2017*), while performing electrophysiological recordings in the striatum (n = 36 mice). C57BL6 mice were used to perform the rest of the electrophysiological recordings of cortical neurons (n = 9).

## Electrophysiological recordings

Adult mice of both sexes (47 animals, 20 males, and 27 females), between 12 and 44 weeks of age were used to perform the experiments (*Supplementary file 1B*). Anesthesia was induced by intraperitoneal injection of ketamine (75 mg/kg) and medetomidine (1 mg/kg) diluted in 0.9% NaCl. A maintaining dose of ketamine (30 mg/kg i.m.) was administrated every 2 hr, after changes in the frequency of spontaneous activity recorded by cortical LFP or reflex responses to paw pinches. Tracheotomy was performed to increase mechanical stability during recordings by decreasing breathing related movements. Mice were placed in a stereotaxic device and air enriched with oxygen was delivered through a thin tube placed 1 cm from the tracheal cannula. Core temperature was monitored and maintained at 36.5 ± 0.5°C using a feedback-controlled heating pad (FHC Inc). Craniotomies were drilled (S210, Camo) at seven sites for patch-clamp and extracellular recordings (from Bregma): AP 0 mm, L 2.5 mm (DMS); AP 0 mm, L 4 mm (DLS); AP 0 mm, L 3,25 mm (DCS); AP 2.7 mm, L 1 mm (FrA); AP 1.5 mm, L 2.0 mm (M1); AP −1.5 mm, L 3.25 mm (S1); AP −3.5 mm from Bregma, L 2.5 mm (V1) (following *Paxinos and Franklin, 2001*). When additional paired recordings were performed, the craniotomies were drilled at the following coordinates (from Bregma): AP 0 mm, L 1 mm (DMS); AP 0 mm, L 4 mm (DLS). Animals were sacrificed after recordings by injecting an overdose of sodium pentobarbital (200 mg/kg I.P.).

## Whole-cell recordings

Whole-cell recordings were obtained in the DS from DLS, DMS, DCS between 2013 and 2647 µm depth. The number of striatal neurons was 126: DLS = 77, from which direct MSNs = 49, indirect MSNs = 34; DMS n = 91, from which direct MSNs = 65, indirect MSNs = 32; DCS n = 17, from which dDCS = 10, iDCS = 7. Cortical neurons (n = 26) were recorded from layer V in FrA, M1, S1 and V1 at a depth of 675–926 µm. In detail, the number of recorded neurons was: Frontal association cortex (FrA, n = 8), primary motor cortex (M1, n = 6), primary somatosensory cortex (S1, n = 6), and primary visual cortex (n = 6). Additional paired simultaneous whole-cell recordings of MSNs (n = 6 pairs) were obtained from DLS and DMS between 2039 and 2348 µm of depth, by means of two micromanipulators (Luigs and Neumann, MRE/MLE Mini 25). All of them in a perpendicular penetration angle of ~ 30°, except for DCS cells, in which the angle was ~20°. The exposed brain was continuously covered with 0.9% NaCl to prevent drying. Signals were amplified using MultiClamp 700B amplifier (Molecular Devices) and digitized at 20 kHz with a CED acquisition board and Spike two software (Cambridge Electronic Design).

Borosilicate patch pipettes (1B150F-4, WPI), were pulled with a Flaming/Brown micropipette puller P-1000 (Sutter Instruments) and had an initial resistance of 6–12 MΩ. Pipettes were back-filled with intracellular solution containing: 125 mM K-gluconate, 10 mM KCl, 10 mM Na-Phosphocreatine, 10 mM HEPES, 4 mM ATP-Mg and 0.3 mM GTP-Na. pH and osmolality were adjusted to ~7.4 and ~280 mOsm/L respectively. Biocytin (0.2–0.4%, Sigma Aldrich) was then added to the intracellular solution to perform cell reconstruction after the experiment. To perform the analysis, 100 s of spontaneous activity (no current injection, no stimulation) were used from the recording. Input resistance (*Table 1*) was measured as the slope of a linear fit between injected depolarizing and hyperpolarizing current steps and membrane potential. Also, in order to improve the quantification of the described inward membrane rectification at hyperpolarized values of the MSNs membrane potentials, mean resistance in response to the negative and positive steps delivered at Up and Down states was also analyzed (*Table 1*), as in *Reig and Silberberg, 2014*. Membrane time constant (tau) was computed as the time to reach 63% of the voltage increment in response to a current pulse. Capacitance was computed as the time constant divided by the input resistance.

Neurons were identified according to their recorded electrical properties and following morphological staining, according to the aspiny dendrites in the case of the ChI and FS interneurons. FS interneurons displayed narrow action potentials, relatively depolarized resting membrane potential, high discharge rate of action potentials, and no apparent inward rectification. Cholinergic interneurons were characterized by their depolarized membrane potential, voltage sag response to current step injections and spontaneous tonic discharge. From the whole population of recorded neurons, 2 of them were identify as FS, one in the DLS and other in the DMS, 3 ChI were recorded in the DMS. These five striatal interneurons were excluded from the data set. Neurons having a deviation by more than 10 mV from their initial resting membrane potential were excluded from analysis. Only MSNs and cortical neurons were included in the data set.

## Extracellular recordings

Extracellular recordings were obtained using unipolar tungsten electrodes with impedance of 1 to 2 MΩ. The electrodes were placed in infragranular layers (~1000 µM depth from the pia) of two cortical regions in each experiment, from FrA, M1, S1, V1 cortex with an angle between 15° and 25°. Recordings were amplified using a differential AC Amplifier model 1700 (A-M Systems) and digitized at 20 KHz with CED and Spike-2, simultaneously with the whole-cell recording.

## Visual stimulation

Visual stimuli were delivered by a white LED positioned 50 mm from the contralateral eye. Duration was 15 ms and it was triggered every 5 s (0.2 Hz) during whole-cell in 17 MSNs recorded in the DCS coordinate and in 10 MSNs recorded in the DMS coordinate. The eye was covered with artificial eye drops (Viscotears, Bausch+Lomb, Germany) in order to prevent drying.

## Optogenetic identification of in vivo recorded neurons

In order to identify 'on line' the specific type of MSNs belonging to the direct (dMSNs) and indirect (iMSNs) pathways, the optopatcher was used (*Katz et al., 2019*; *Katz et al., 2013*; *Ketzef et al.,*

*2017*; A-M systems, WA USA). Controlled pulses (SLA-1000–2, two channel universal LED driver, Mightex systems) of blue light (Fibre-coupled LED light source FCS-0470–000, 470 nm, Mightex systems) through Spike two software were delivered using an optic fibre (200 µm diameter, handmade) inserted into the patch-pipette, while recording their spontaneous activity (*Figure 7A*). One or two serial pulses with five light steps of 500 ms each were delivered every 2 s with increasing intensity from 20% to 100% of full LED power (minimal light intensity was 0.166 mW, maximal intensity was 0.83 mW measured at the tip of the fibre). Power light was measured with an energy meter console (PM100D, Thorlabs). Positive cells responded to light pulses by depolarizing the membrane potential (*Figure 7A*, upper trace), responding within 2.69 ± 1.37 ms (ranging from 0.8 to 5 ms) to light pulses by a step-like depolarization of 13.3 ± 8.28 mV (ranging from 4 to 33 mV), therefore were classified as indirect MSNs. Negative cells did not show any depolarization to light pulses (*Figure 7A*, bottom trace) and were classified as a putative direct MSNs.

## Morphological reconstruction
At the end of each experiment, mice were sacrificed with a lethal dose of sodium pentobarbital and perfused with a solution containing 4% paraformaldehyde in 0.1 M phosphate buffer (PB, pH 7.4). Brains were extracted and stored in PBS solution until the cutting. Before cutting, brains were transferred into PBS containing 30% sucrose for at least 48 hr. Coronal slices (25 µm thick) containing the entire striatum from the recorded side (from AP 1.4 mm to AP −1.3 mm, following *Paxinos and Franklin, 2001*), were obtained using a digital automatic cryotome and collected on gelatine coated slides. Sections were incubated over night with Cy3-conjugated streptavidin (Jackson Immuno Research Laboratories) diluted (1:1000) in 1% BSA, 0.3% Triton-X 100 in 0.1 M PBS. Finally, the glass slides were covered with mowiol (Calbiochem) and imaged. Neurons were then reconstructed using a fluorescence microscope (DM 6000B, Leica) and a camera (DC350 FX, Leica) and then processed by ImageJ.

## Extraction of up and down states
In order to isolate Up and Down states from the membrane voltage we started by smoothing the trace using a 200 ms window. Then, we extracted the Up states using a threshold consisting of the mean value of the membrane potential (Vm) plus 0.5 standard deviations. Then, we merged the parts of fragmented Up states, detected as an interval shorter than 250 ms between the transition of a prospective Up state to a prospective Down state to the transition from a prospective Down state to a prospective Up state. Finally, detected Up states shorter than 200 ms where discarded.

When studying propagation of Up states, we considered two Up states from two different cortical regions, or a cortical LFP recording and an MSN to be part of the same SWO, thus propagating from one region to the other, when their onsets were closer in time than 500 ms.

## Feature extraction of Up states
In order to describe the properties of the SWO of the recorded MSNs, we computed 13 parameters (*Figure 1D*). First, we subtracted the IMFs carrying oscillations faster than 50 Hz from the MSNs membrane voltage recordings to eliminate the spikes. Then, we divided the trace into Up and Down states and calculated the different features; the mean value of the membrane potential in the Up state (Feature number 1 (1)), the standard deviation of the Up state membrane potential (2), minimum (most hyperpolarized value) membrane potential in the Up state (3) and maximum (most depolarized value) membrane potential in the up state (4). Additionally, we computed the peak to peak distance (12), which is the difference between the most depolarized and most hyperpolarized values of the membrane potential of the Up state, the maximum (6) and minimum (7) value of the derivative of the values of the membrane potential in the Up state, as well as the number of peaks in the Up state (11), understood as big, slow changes in the membrane potential inside the Up state, which created diplets or triplets during the Up state (*Figure 4A,B*). To do so, we used the *findpeaks* Matlab function on the smoothed (200 ms smooth) Up state with a minimum peak height of 0.3 standard deviations and a minimum distance between peaks of 160 ms. We also computed the length of the Up state (13), the mean amplitude of the Up state, computed as the difference in membrane potential average value between Up and Down states (5) and the speed of the transition from the Down to the Up state (8), and from the Up to the Down state (9) as well as the Slope transition ratio,

computed as the speed of transition from Down to Up states divided by the one from Up to Down states (10). In order to measure the slope speeds, we used a vector consisting in −100 to +200 ms (300 ms total) relative to the transition either to the Up or the Down state. Then, we smoothed (100 ms smooth) the selected region and computed its derivative. We delimited the slope computing the region over a threshold calculated as the mean plus 0.3 standard deviations of the derivative. Once we had delimited the transition, we fitted a lineal function to that region of the recording to compute the slope speed.

## Detection of intracellular synaptic events

Intracellular synaptic currents were extracted using sharp deflections of the membrane voltage using a modification of the method presented here (*Compte et al., 2008*). In brief, we applied a Parks-McClellan low-pass differentiator filter (order 20, cut off at 200 Hz) and computed the first derivative of the intracellular membrane potential. Then, we extracted the temporal profile of depolarizing and hyperpolarizing events to the recorded cell using a threshold of the first derivative. We empirically determined a two standard deviations threshold to extract them. Next, we aligned the detected membrane deflections for both the hyperpolarizing and depolarizing events to the phase of the SWO as computed using NA-MEMD and normalized them to their respective maximum (*Figure 4E*) in order to study the profile of the recruitment of excitation and inhibition during SWO. We computed the depolarizing/hyperpolarizing ratio subtracting the hyperpolarizing normalized value to the depolarizing one at each phase point.

For our whole-cell recordings, we have used a low-chloride intracellular solution (described above), in which the reversal potential for excitation is ~ −5 mV and $GABA_A$ inhibition ~ −70 mV. The average membrane potential during the Down states of the recorded MSNs was −69.2 ± 6.48 mV for DLS-MSNs and −69.31 ± 7.68 mV for DMS-MSNs. Around these voltages, the membrane fluctuations are depolarizing and excitatory. However, during the Up states (mean voltage values: −57.24 ± 6.34 mV for DLS-MSNs and −52.95 ± 7.08 mV for DMS-MSNs), both depolarized and hyperpolarized events can be detected (*Figure 4D*). As previously discussed (*Compte et al., 2008*), this method does not extract all the EPSPs or IPSPs of the recorded cell, but it provides a major picture of the convergence of coordinated excitatory or inhibitory inputs onto the recorded cell that produced sharp deflections in their membrane potential. Therefore, this method is sufficient to study the temporal profiles of hyper/depolarizing events onto the recorded cells during SWO.

## Lateromedial classification

We classified MSNs from DLS and DMS using a Support Vector Machine (SVM) (*Cortes and Vapnik, 1995*) with a linear kernel. This supervised classification method estimates a hyperplane to separate both populations. Classification was cross-validated 10 times.

In order to determine which features were more relevant for the classification of DLS- and DMS-MSNs, we used Recursive Feature Elimination. This approach organizes the features relative to their importance to the classification by recursively pruning the least relevant features starting from the initial set.

## MSNs distribution along the classification boundary

Support Vector Machines (SVMs) are supervised classifiers that predict the category of a new entry based on their previous training history and are not suitable to predict new categories in novel data. In order to study the spontaneous activity of DCS-MSNs, we examined its distribution in the classification space, relative to the classification boundary instead of directly using the output of a classifier. This approach provides information, not only about whether the new data (DCS-MSNs) creates a new cluster in the classification space, but also about where is it placed along the classification axis used to discriminate between DLS- and DMS-MSNs. Hence, it provides a more informative approach to the distribution of DCS-MSNs relative to DLS- and DMS-MSNs than a direct multi-category classification (i.e. discriminate versus Hypotheses 1 and 2, see Results).

To calculate the distribution of the data along the classification boundary, we first obtained the coefficients and intersection values of the SVM used to classify the MSNs belonging to DLS and DMS. Once we obtained these parameters, we used them to project all data points to this hypervector and computed the probability density function of the new distributions (*Figure 3D*). Now, the

data was distributed along a single hypervector and we could compute the distribution of the MSNs along the classification boundary in a comprehensible manner to study the distribution of the DCS-MSNs.

## SWO computation using NA-MEMD

We used Noise-assisted Multivariate Empirical Mode Decomposition (NA-MEMD) algorithm (*ur Rehman and Mandic, 2011*) together with Hilbert transform (*Huang et al., 1998*) for the analysis of high-frequency oscillations both in LFPs and MSNS membrane potential, as well as the cortico-striatal propagation of the SWO. Because neuronal oscillations are characterized by nonlinear properties, this algorithm is suited to decompose nonlinear nonstationary signals (*Alegre-Cortés et al., 2017*; *Alegre-Cortés et al., 2016*; *Hu and Liang, 2014*; *Mandic et al., 2013*). The original EMD (*Huang et al., 1998*) is a data-driven algorithm suitable for nonlinear and non-stationary signals that does not rely on any predetermined template. It decomposes a given signal into a subset of oscillatory modes called Intrinsic Mode Functions (IMFs) (*Figure 2—figure supplement 2*). Each IMF contains the oscillations of the original data in a certain frequency range. Then, Hilbert transform is applied onto each IMF in order to compute its instantaneous frequency and amplitude. The MEMD (*Rehman and Mandic, 2010*) is a multivariate extension of the original EMD to n-dimensional signals. The MEMD is computed simultaneously in all dimensions of the signal to ensure the same number of IMFs as output. In addition, new dimensions can be added to the data containing White Gaussian Noise (WGN) to increase its performance, as it has been described that WGN addition reduces mode mixing produced by signal intermittence (*Wu and Huang, 2009*), acting as a quasi-dyadic filter that enhances time frequency resolution (*Flandrin et al., 2004*; *ur Rehman and Mandic, 2011*). The application of MEMD to the desired signal together with extra White Gaussian Noise dimensions is known as NA-MEMD analysis (*ur Rehman and Mandic, 2011*). EMDs algorithms plus Hilbert Transform have been increasingly used in neuroscience during the last years as they produce an enriched Time-Frequency (T-F) Spectrum when compared to traditional T-F linear tools (*Alegre-Cortés et al., 2016*; *Hu and Liang, 2014*). In our work, we applied NA-MEMD algorithm to a multivariate signal composed by the intracellular recording, both LFPs and one extra WGN channel. By means of this analysis, we could warrant that the number of IMFs was the same in the intracellular and the LFPs recordings for a direct comparison of the SWO extracted using this method. In order to apply NA-MEMD analysis to our data, we adapted MEMD Matlab package (http://www.commsp.ee.ic.ac.uk/mandic/research/emd.htm). Standard stopping criterion was described elsewhere (*Rilling et al., 2003*). At last, we extracted the IMF carrying the SWO as the one with maximum correlation with the membrane voltage and visually confirmed it in all the recorded cells (*Figure 1C*). Once we isolated the SWO of each recording using NA-MEMD, they were stored for further analysis.

## Hilbert transform

We computed the frequency of the SWO, and theta (6–10 Hz), beta (10–20 Hz) and gamma (20–80 Hz) bands as the instantaneous frequency using the Hilbert transform (*Huang et al., 1998*). For a given time series x(t), its Hilbert transform H(x)(t) is defined as:

$$d(x)(t) = \frac{1}{\pi} C \int_{-\infty}^{\infty} \frac{x(t́)}{t - t́} dt́$$

where C indicates the Cauchy principal value. Hilbert transform results in a complex sequence with a real part which is the original data and an imaginary part which is a version of the original data with a 90° phase shift; this analytic signal is useful to calculate instantaneous amplitude and frequency; instantaneous amplitude is the amplitude of H(x)(t), instantaneous frequency is the time rate of change of the instantaneous phase angle.

## Statistical analysis

All experimental comparisons were tested using the Wilcoxon Rank-sum test except stated otherwise. Directionality of the SWO in the cortex and in the striatum was tested using Wilcoxon signed rank test. Phase locking of different frequency bands was tested using Rayleigh test. When required,

alpha values for multiple comparisons were corrected using Holm-Bonferroni correction. Error bars presented in the graphs represent the standard deviation unless stated otherwise.

## Acknowledgements

We thank Gilad Silberberg (Karolinska Institutet) for generous donation of D2-Cre(ChR2)-YFP mouse and for helping us to start-up the laboratory. We also want to thank Gilad Silberberg, Juan Lerma, Liset M de la Prida, Sandra Jurado, John F Wesseling, Maya Ketzef, Cristina García-Frigola, Juan Pérez-Fernández and Roberto de la Torre for critical reading and discussions on this. J A-C and M S are supported by the CSIC-Severo Ochoa Excellence Programme of the Instituto de Neurociencias [SEV2013-0317]; R M is supported by La Caixa-Severo Ochoa [2016/00006/001]. The study was supported by MINECO Fellowship [BES-2015–072187], Starting Grant I+D Jovenes Investigadores [BFU2014-60809-IN] and CSIC-Severo Ochoa Excellence Programmes of the Instituto de Neurociencias [SEV-2013–0317 and SEV-2017–0723].

## Additional information

### Funding

| Funder | Grant reference number | Author |
|---|---|---|
| Ministerio de Economía, Industria y Competitividad, Gobierno de España | BFU2014-60809-IN | Ramon Reig |
| Ministerio de Economía, Industria y Competitividad, Gobierno de España | SEV-2013-0317 and SEV-2017-0723 | Ramon Reig |
| Ministerio de Economía, Industria y Competitividad, Gobierno de España | SEV2013-0317 | María Sáez |
| la Caixa Foundation | 2016/00006/001 | Roberto Montanari |
| MINECO | BES-2015–072187 | Ramon Reig |

The funders had no role in study design, data collection and interpretation, or the decision to submit the work for publication.

### Author contributions

Javier Alegre-Cortés, Formal analysis, Investigation, Visualization, Writing - original draft, Writing - review and editing; María Sáez, Formal analysis, Investigation, Visualization, Methodology, Writing - review and editing; Roberto Montanari, Investigation, Methodology, Writing - review and editing; Ramon Reig, Conceptualization, Resources, Data curation, Supervision, Funding acquisition, Investigation, Methodology, Writing - original draft, Project administration, Writing - review and editing

### Author ORCIDs

Javier Alegre-Cortés (iD) https://orcid.org/0000-0003-0888-7542
María Sáez (iD) https://orcid.org/0000-0001-9137-6692
Roberto Montanari (iD) https://orcid.org/0000-0002-4331-8460
Ramon Reig (iD) https://orcid.org/0000-0002-6475-4181

### Ethics

Animal experimentation: All experimental procedures conformed to the directive 2010/63/EU of the European Parliament and the RD 53/2013 Spanish regulation on the protection of animals use for scientific purposes, approved by the government of the Autonomous Community of Valencia, under the supervision of the Consejo Superior de Investigaciones Científicas and the Miguel Hernandez University Committee for Animal use in the Laboratory.

Decision letter and Author response
Decision letter https://doi.org/10.7554/eLife.60580.sa1
Author response https://doi.org/10.7554/eLife.60580.sa2

## Additional files

### Supplementary files

• Supplementary file 1. Additional data. (A) Featurization of the SWO of DCS-MSNs. Note that the number preceding the feature labels are the same as the ones showed in *Figure 3*. All values display means ± standard deviation. (B) Data set description. The three first rows are total values; the rest display the mean ± standard deviation.

• Transparent reporting form

### Data availability

All data generated during and/or analysed during the current study, as well as the required code to reproduce the figures, is available on the CSIC public repository. This is the URL access https://digital.csic.es/handle/10261/229794.

The following dataset was generated:

| Author(s) | Year | Dataset title | Dataset URL | Database and Identifier |
|---|---|---|---|---|
| Javier A-C, Sáez M, Montanari R, Reig R | 2021 | Medium spiny neurons activity reveals the discrete segregation of mouse dorsal striatum | https://doi.org/10.20350/digitalCSIC/13750 | DIGITAL.CSIC, 10.20350/digitalCSIC/13750 |

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
