## [Decision Letter]

**Acceptance summary:**

Using multiple in vivo patch-clamp recordings coupled to local field potentials recordings in anesthetized animals, Alegre-Cortes et al. have explored and compared the properties of Dorso-medial striatum and Dorso-Lateral Striatum circuits. The technical approaches to explore the question are particularly impressive as double in vivo patch-clamp recordings in striatum have never been performed before. It allowed comparing the two striatal circuits and/or cortical neuronal responses simultaneously. The study brings new insights on the distinction between functional areas within the striatum using slow wave oscillations recordings and visual stimulations. The results are interesting and convincing, and the methodology appropriate to draw the conclusions presented here. The revised manuscript shows large improvements in linking different technical aspects of the study and the detailed explanations increase the clarity of the conclusions. The main conclusion (and novelty) is that the difference between the two striatal regions is not solely based on their cortical inputs but also on different properties of MSNs belonging to each region. Together these new findings change the current view of striatal circuits.

**Decision letter after peer review:**

Thank you for submitting your article "Medium spiny neurons activity reveals the discrete segregation of mouse dorsal striatum" for consideration by *eLife*. Your article has been reviewed by two peer reviewers, and the evaluation has been overseen by a Reviewing Editor and Laura Colgin as the Senior Editor. The following individual involved in review of your submission has agreed to reveal their identity: John Reynolds (Reviewer #2).

The reviewers have discussed the reviews with one another and the Reviewing Editor has drafted this decision to help you prepare a revised submission.

Summary:

Using double in vivo patch-clamp recordings in striatum the authors have compared DMS and DLS striatal and/or cortical neuronal responses simultaneously. The study brings sheds new light on the existence of distinct functional areas within the striatum using slow wave oscillations recordings and visual stimulations. The main conclusion is that the difference between the two striatal regions are not solely based on their cortical inputs but also on different properties of MSNs belonging to each region.

Revisions:

1) The authors must provide the rationale for their choice of the parameters chosen to distinguish DMS from DLS neurons. First, if it is not sufficiently for expert reviewers, "average" readers will be at lost. Second, the whole paper relies on these parameters and careful explanations as to why and how is warranted.

2) Double patch in vivo is "heroic" and while the reason for performing the study in anesthetized animals might be obvious to most, the Discussion must be expanded. In particular, slow wave oscillations are a specific regime within these networks and analyzing the networks in that state might be limiting. In this context, Mahon et al., 2006, showed a very different activity in the corticostriatal oscillations between SWO and awake animals. The results of the present study should be discussed in light of the different regimes existing in the corticostriatal pathways depending on the vigilance state of the animals.

3) Concerning the propagation of SWO in the striatum, the Authors describe no difference in correlation with S1 inputs between DMS and DLS. How do they explain this? These experiments show nicely the sequence of activation in the corticostriatal loops. Nevertheless, they could argue against the conclusion of the study. If there is a difference in correlation between cortical areas projecting to only one striatal region (DMS or DLS) and no difference in correlation with S1, projecting to both, this would argue that the cortical inputs are the most important to drive DMS or DLS activity, and not the local circuits. This should be discussed.

4) A more detailed methodological explanation of how the Noise-assisted multivariate empirical mode decomposition (NA-MEMD) was implemented to analyze membrane oscillations of single neurons as well as a graphical explanation as to how the potentials have been derived are warranted.

Reviewer #1:

Using multiple in vivo patch-clamp recordings coupled to LFP recordings in anesthetized animals, the study by Alegre-Cortes et al. explores the difference between DMS and DLS circuit properties. The technical approaches to explore the question are impressive as double in vivo patch-clamp recordings in striatum have never been performed before and allows to compare DMS and DLS striatal and/or cortical neuronal responses simultaneously. The study brings new insights on the distinction between functional areas within the striatum using slow wave oscillations recordings and visual stimulations. The main conclusion is that the difference between the two striatal regions are not solely based on their cortical inputs but also on different properties of MSNs belonging to each region.

The results are interesting and convincing, and the methodology appropriate to draw the conclusions presented here. Nevertheless, attempts to link different aspects of the study and more detailed explanations would improve the clarity of the conclusions of the manuscript. Please see below some specific points.

1) In its present form the manuscript presents a lot of different parameters to show differences in DMS and DLS. It would be important to link the different properties together to clearly highlight which components would be the most important to distinguish DMS or DLS and therefore draw a clearer conclusion about their different properties.

For example, data regarding the electrophysiological properties of DMS- and DLS-MSNs would be better on the main figures and the authors should try to correlate these properties and the up states characteristics. It would be useful to explain better the rationale of the choice of these 3 specific parameters and try to include others in the overall interpretation.

Another example, linking Figure 1 and Figure 4 would show relationship between different parameters of cortical oscillations and related ones in DMS and DLS. Figure 4: Why only focusing on the number of peaks during the up states while interesting differences have been highlighted in Figure1 on other features of the up states? The reviewer would recommend to include analysis of up state amplitude and slopes in this part of the manuscript.

2) Concerning the propagation of SWO in the striatum, the Authors describe no difference in correlation with S1 inputs between DMS and DLS. How do they explain this? These experiments show nicely the sequence of activation in the corticostriatal loops. Nevertheless, they could argue against the conclusion of the study. If there is a difference in correlation between cortical areas projecting to only one striatal region (DMS or DLS) and no difference in correlation with S1, projecting to both, this would argue that the cortical inputs are the most important to drive DMS or DLS activity, and not the local circuits. This should be discussed.

3) Related to this, sensory stimulation data did not support the conclusions of a clear segregation of DMS and DLS properties. The explanation for such a difference between sensory stimulation and SWO should be discussed in more details.

The rationale for using visual (as opposed to whisker) stimulation is that it targets only the medial part of the striatum. This allowed the Authors to show a distinction between DMS and DCS, but, which was mainly due to different cortical input origin. Moreover, since the aim of the study is to highlight that differences within the striatum that are not due to different inputs, maybe using a common cortical input to activate both striatal networks would probably be suitable to explore if specific responses are only due to local properties. Whisker stimulation was excluded to test the distinction between DMS and DLS because S1 projects to both regions, but is it because the weight of S1 inputs are different on DMS and DLS and so it would be impossible to dissociate inputs from local circuits? If possible a common equivalent input could be used for the aim the study. These experiments would not be required for the present study but maybe the Discussion should include these points.

4) Concerning the measure of the depolarizing/hyperpolarizing ratio (DH ratio), it is not clear how the authors conclude of striatal microcircuits contribution from the recordings in Figure 4 and how it can be isolated from cortical drive from the recordings? If not clearly stated, the reviewer is not sure that this analysis is necessary to support the author's conclusions.

5) The method used to do the delineation of the DCS is not clear and should be better explained in the main text.

6) Slow wave oscillations are a specific regime within these networks and even though informative, analyzing the networks in that state might be limiting. Mahon et al., 2006, they show a very different activity in the corticostriatal oscillations between SWO and awake animals. The results of the present study should be discussed in light of the different regimes existing in the corticostriatal pathways depending on the vigilance state of the animals.

7) One last suggestion is that direct and indirect pathway data could be presented at the beginning with intrinsic properties and up states characterization of the MSNs in DMS and DLS to avoid coming back to intrinsic properties at the end of the manuscript.

Reviewer #2:

This interesting study presents a number of analyses of in vivo intracellular recordings of the membrane potential activity of MSNs in the striatum, which indicate that there are differences in electrophysiological properties in dorsomedial vs dorsolateral striatum. Furthermore there appears to be a discrete demarcation between neurons in each area with only a small overlap and no gradient. The authors then go on to record from afferent areas to determine if the patterns of activity in the MSNs are echoed by differences in neuronal activity in functional areas projecting to the striatum. Finally, they further demarcate the neurons in each area by whether they are dMSNs or iMSNs, and find that there were further differences of neurons in the lateral striatum only.

There are a number of strengths to this paper, primarily by virtue of the very difficult experiments that these involve – of particular note is the heroic dual patching experiments which deserve special mention – there have been only a couple of dual intracellular in vivo studies before but in rats using sharp electrodes – obtaining low access resistance recordings from simultaneously recorded neurons using patch recording in mice is very notable. Overall the conclusions are robust and these raise the possibility of why these differences are present (cell intrinsic vs afferent properties vs local neuromodulation), which the authors speculate on but do not investigate.

Although I can understand how temporal measures such a durations of states and trajectory of transitions can be quantified using the NA-MEMD algorithm, I cannot see how the actual average and SD of the up and down state membrane potentials have been extracted from the IMFs, when looking at those exemplified in Figure 2—figure supplement 2C and the SWO also shown in Figure 1C. Was there an additional step needed to calculate a probability density function from the distribution of the membrane potentials, as in Figure 5C? If not, because as the authors say it is the first time that the algorithm has been used to analyse membrane oscillations of single neurons, a much more detailed methodological graphical explanation as to how the potentials have been derived is warranted.

The authors found "an Up state in FrA preceding an Up state in V1 in the 71% of times (p value< 0.0001), data not shown" – this is itself very interesting and the data SHOULD be shown as further suggestive (correlative) evidence that the propagation of Up state timing from lateral to medial striatum is driven by frontal to visual cortex transition of slow wave.

The observation of a difference between cellular properties and SWO measures in lateral and medial regions is just that, without any expose of the underlying reason. This paper could be much more enriched by some investigation of a mechanism that might explain these differences. In lieu of a full morphological comparison, one possibility the authors do raise that is quite achievable in a relatively short time is considering the effects of differences in dopamine innervation, either directly or indirectly via cholinergic neuronal innervation. Indeed there is known to be medial to lateral gradients of many neuromodulatory components including dopamine receptors which might contribute to the regional differences. The authors should consider undertaking a series of comparison experiments under blockade of D1 and D2 antagonists (separately and together, or 6-OHDA depletion better still for the latter) and see if this normalises the observed differences between neurons. This is relatively more straightforward than trying to normalise FSI innervation and in my opinion is much more likely to provide at least a partial explanation.

---

## [Author Response]

Revisions:1) The authors must provide the rationale for their choice of the parameters chosen to distinguish DMS from DLS neurons. First, if it is not sufficiently for expert reviewers, "average" readers will be at lost. Second, the whole paper relies on these parameters and careful explanations as to why and how is warranted.

We have now included more detailed and careful explanations of the 13 features of the SWO in the Materials and methods section. In addition, an explanation about the 3 selected features by the Recursive Feature Elimination analysis has been included, as the ones that their combination better discriminate between de DLS- and DMS-MSNs (number of peaks in the Up states, amplitude of the Up states and the Up to Down transition slopes), in the Result section.

2) Double patch in vivo is "heroic" and while the reason for performing the study in anesthetized animals might be obvious to most, the Discussion must be expanded. In particular, slow wave oscillations are a specific regime within these networks and analyzing the networks in that state might be limiting. In this context, Mahon et al., 2006, showed a very different activity in the corticostriatal oscillations between SWO and awake animals. The results of the present study should be discussed in light of the different regimes existing in the corticostriatal pathways depending on the vigilance state of the animals.

We have discussed this important idea in the context of Mahon et al., 2006 (Discussion). We also answer to reviewer 1 in detail below.

3) Concerning the propagation of SWO in the striatum, the Authors describe no difference in correlation with S1 inputs between DMS and DLS. How do they explain this? These experiments show nicely the sequence of activation in the corticostriatal loops. Nevertheless, they could argue against the conclusion of the study. If there is a difference in correlation between cortical areas projecting to only one striatal region (DMS or DLS) and no difference in correlation with S1, projecting to both, this would argue that the cortical inputs are the most important to drive DMS or DLS activity, and not the local circuits. This should be discussed.

This important point is now extensively discussed in the text and also please find our extended response to reviewer #1 (see below).

4) A more detailed methodological explanation of how the Noise-assisted multivariate empirical mode decomposition (NA-MEMD) was implemented to analyze membrane oscillations of single neurons as well as a graphical explanation as to how the potentials have been derived are warranted.

We have clarified the analysis performed during this paper, as sometimes it was ambiguous when the raw traces or an IMF (the output of the NA-MEMD) was being used. It is now clearer when the NA-MEMD is used to analyse membrane oscillations in the Figures 2 and 5. The presence of an IMF in the Figure 1 was particularly confusing. Therefore, we have removed the IMF from panel 1C and now we have added the IMFs to Figure 5A-B, which illustrates better when is used.

Both the Up state properties in Figure 1 and hyperpolarizing and depolarizing potentials in Figure 4 were computed using the raw membrane potential and not the NA-MEMD, as detailed in the answer to Dr Reynolds. This new version of the manuscript states more clearly when the NA-MEMD is applied.

Reviewer #1:Using multiple in vivo patch-clamp recordings coupled to LFP recordings in anesthetized animals, the study by Alegre-Cortes et al. explores the difference between DMS and DLS circuit properties. The technical approaches to explore the question are impressive as double in vivo patch-clamp recordings in striatum have never been performed before and allows to compare DMS and DLS striatal and/or cortical neuronal responses simultaneously. The study brings new insights on the distinction between functional areas within the striatum using slow wave oscillations recordings and visual stimulations. The main conclusion is that the difference between the two striatal regions are not solely based on their cortical inputs but also on different properties of MSNs belonging to each region.The results are interesting and convincing, and the methodology appropriate to draw the conclusions presented here. Nevertheless, attempts to link different aspects of the study and more detailed explanations would improve the clarity of the conclusions of the manuscript. Please see below some specific points.1) In its present form the manuscript presents a lot of different parameters to show differences in DMS and DLS. It would be important to link the different properties together to clearly highlight which components would be the most important to distinguish DMS or DLS and therefore draw a clearer conclusion about their different properties.

We thank reviewer #1 for their supportive comments.

We started the study with a statistical description of the magnitude and shape of the Up states, using 13 different parameters, now extensively explained in the new version (Materials and methods section). The characterization of the Up state features, related with its amplitude, duration and transition slopes have been previously used (Perez-Zabalza et al., 2020; Sanchez-Vives et al., 2010). From the 13 Up state features used, 11 of them were statistically different between DLS and DMS (Figure 1D). Then, we investigated for the combination of features that together were more powerful to distinguish between DLS- and DMS-MSNs. To do so, we recursively pruned the features using a Support Vector Machine; this implies that, starting from the whole set of parameters, they were recursively eliminated depending on their relevance for the classification of DLS- and DMS-MSNs. As a result, we obtained a sorting of the features depending on their significance for the separation of DLS- and DMS-MSNs. The final result consisted in a combination of three parameters, number of peaks (feature nº 11), amplitude (feature nº 5) and Up to Down transition slope (feature nº 9), which when combined let us classify MSNs (Figure 1E).

We have amended the Results section “The spontaneous activity from DLS- and DMS-MSNs have different attributes” in order to clarify the logic behind the analysis.

Figure 1 displays the most relevant information underlying the division of DLS and DMS. This is why we decided to place it at the beginning of the manuscript.

For example, data regarding the electrophysiological properties of DMS- and DLS-MSNs would be better on the main figures and the authors should try to correlate these properties and the up states characteristics. It would be useful to explain better the rationale of the choice of these 3 specific parameters and try to include others in the overall interpretation.

We thank reviewer #1 for this suggestion. We have clarified the logic and technique behind the selection of the three specific parameters.

As we answered in the previous question, the rationale of the choice of the 3 specific features is based on their power to classify between MSNs and is determined by the Recursive Feature Elimination analysis.

Our results support that, while it is clear that the differences between membrane electrophysiological properties of DLS and DMS are relevant for our understanding of the DS and may have a small contribution in the difference of spontaneous activity of each region, the main agents modulating the SWO in DLS and DMS are the cortical excitatory inputs and the local striatal interactions (which includes different gradients of neuromodulators and interneurons). Thus, we do not argue for a direct and measurable correlation between any membrane property and Up state feature.

In an effort to clarify this question, we computed the correlation of the membrane properties and the Up state features of the MSNs. Nevertheless, no comparison had an absolute correlation value over 0.3 (Figure 1—figure supplement 2), suggesting that there is not a lineal relationship between these two groups of properties (this result is now added in the text). For instance, based on the electrophysiological properties, it is expected that higher input resistance will contribute to generate higher amplitudes when neurons are integrating synaptic inputs, but it was not the case. Up states amplitudes were higher in the DMS-MSNs with smaller values of input resistance than DLS-MSNs (Figure 1D, features nº 4 and 5, and Table 1).

Thus, after carefully discussing the suggestion of reviewer #1, we think that the actual distribution of figures and tables is the best way to explain our results.

Another example, linking Figure 1 and Figure 4 would show relationship between different parameters of cortical oscillations and related ones in DMS and DLS. Figure 4: Why only focusing on the number of peaks during the up states while interesting differences have been highlighted in Figure1 on other features of the up states? The reviewer would recommend to include analysis of up state amplitude and slopes in this part of the manuscript.

We have modified the Results section “DLS- and DMS-MSNs segregation is explained by their membrane potential transitions” to clarify the question of the reviewer. This section and Figure 4 were done with the aim to understand the main features of the Up states selected by the Recursive Features Elimination analysis classifying MSNs (Figure 1E). We have also extended the Discussion section “DLS and DMS are two different functional circuits in mouse” in order to clarify this question.

Figure 4D-F describes the differences in slopes. We have made an effort to improve the visualization of the DH ratio and its relation with the slopes, underlying changes between DLS and DMS-MSNs; this analysis suggests that the balance of excitatory and inhibitory inputs onto the MSNs is different during the transitions between states, suggesting a different interaction between the long range excitatory inputs and the local inhibitory ones. We have also added a new panel showing the grand average of the upward and downward states transitions slopes (new Figure 4F).

The third relevant feature to classify MSNs was the Up state amplitude (Figure 1E, feature nº 5). In order to explain the Up state amplitude disparity, our initial approach was to calculate the MSNs input resistance, because higher input resistance should increase the amplitude of the synaptic inputs. However, against this hypothesis, amplitude was higher in DMS-MSNs. The number of direct excitatory inputs that single DMS-MSNs receives from cortex could be a candidate underlying this unexpected result. It was described that the medial region of the dorsal striatum is innervated by the highest degree of cortical input heterogeneity (Hunnicutt et al., 2016). Thus, it could be possible that single DMS-MSNs receive larger number of cortical inputs which promote bigger Up states amplitudes, even with smaller input resistance.

2) Concerning the propagation of SWO in the striatum, the Authors describe no difference in correlation with S1 inputs between DMS and DLS. How do they explain this? These experiments show nicely the sequence of activation in the corticostriatal loops. Nevertheless, they could argue against the conclusion of the study. If there is a difference in correlation between cortical areas projecting to only one striatal region (DMS or DLS) and no difference in correlation with S1, projecting to both, this would argue that the cortical inputs are the most important to drive DMS or DLS activity, and not the local circuits. This should be discussed.

We thank the reviewer #1 to help us to clarify this important question. The general conclusion extracted from our results claims that the entire distribution of corticostriatal projections together with the local microcircuit properties are both necessary for the functional division of the DLS and DMS. Also, it is important to remember that the striatum is basically a GABAergic network that requires excitatory inputs acting as activity driver.

The approach used in Figure 5 aims to understand the global correlation of the striatal SWO with respect to different cortical areas, which is different from the modulation of activity once the MSNs are in the active Up state (inside the up state). In order to understand the propagation of the cortical SWO to the striatum we used the IMF that carried this wave (~1Hz), this method is now illustrated in Figure 5A-B. Note how this function retains the Up-Down dynamics but excludes the fast modulation of the membrane voltage. Thus, the cross-correlation analysis showed in Figure 5 gave us information about which regions of the cortex were the ones “triggering” the Up states to DLS and DMS.

Following the indications of reviewer #1, we have extended the Discussion of the paper to address this concerns, adding a new reference (Charpier et al., 2020).

3) Related to this, sensory stimulation data did not support the conclusions of a clear segregation of DMS and DLS properties. The explanation for such a difference between sensory stimulation and SWO should be discussed in more details.The rationale for using visual (as opposed to whisker) stimulation is that it targets only the medial part of the striatum. This allowed the Authors to show a distinction between DMS and DCS, but, which was mainly due to different cortical input origin. Moreover, since the aim of the study is to highlight that differences within the striatum that are not due to different inputs, maybe using a common cortical input to activate both striatal networks would probably be suitable to explore if specific responses are only due to local properties. Whisker stimulation was excluded to test the distinction between DMS and DLS because S1 projects to both regions, but is it because the weight of S1 inputs are different on DMS and DLS and so it would be impossible to dissociate inputs from local circuits? If possible a common equivalent input could be used for the aim the study. These experiments would not be required for the present study but maybe the Discussion should include these points.

We thank reviewer #1 for this input. We have extended the Discussion section entitled “DLS and DMS are not defined by their sensory responses”, to include it.

DMS-MSNs, as well as DLS-MSNs respond to whisker (tactile) stimulation with the same onset delay, most probably because both regions share the axonal projections from S1. This corticostriatal innervation is characterized by higher axonal density in the lateral pole of the DS and lower density in its medial pole (Figure Supplementary 1 in Reig and Silberberg 2014). Moreover, previous studies described a gradient of S1 axons invading both regions (Alloway et al., 2006; Brown et al., 1996), with progressive loss of axons from lateral to medial territories and without any anatomical border limiting this cortical innervation. Therefore, it is expected that this gradual decrease of projections from lateral to medial striatum will be reflected in progressive changes on the spontaneous SWO of MSNs as well, impeding the localization of a functional border between DLS and DMS.

On the other hand, we showed that visual responses display very different properties in DCS and DMS coordinates (Figure 3 F-I), including their onset delays. Despite these significant differences, they were not suitable to discriminate the border between DLS and DMS. This is something that occurs with a large number of results showed in our study; for instance, DLS- and DMS-MSNs differ in their intrinsic electrophysiological properties, direct and indirect pathways properties, sensory responses and highfrequency oscillatory activity. Even these results are essential to describe MSNs properties in both striatal regions, most of them do not contain enough information to discriminate between DLS and DMS circuits, based on our classifier. In contrast, we found that the combination of few Up states features of the SWO are enough to identify between striatal regions with ~90% of accuracy. All of these reinforce the idea of the SWO as a reliable method to study functional connectivity.

4) Concerning the measure of the depolarizing/hyperpolarizing ratio (DH ratio), it is not clear how the authors conclude of striatal microcircuits contribution from the recordings in Figure 4 and how it can be isolated from cortical drive from the recordings? If not clearly stated, the reviewer is not sure that this analysis is necessary to support the author's conclusions.

Our DH ratio provides information to characterize the Up states transition slopes, one of the Up state features selected to identify between striatal regions. In particular, the divergence in the DH ratio between DLS and DMS-MSNs implies that the balance between depolarizing and hyperpolarizing inputs is different during these transitions. We found that DMS-MSNs have larger ratio of depolarizing events during upward transition and larger proportion of hyperpolarizing events during downward transition with respect to the DLS-MSNs (Figure 4E). These results supports our suggestion of the role of striatal interneurons as one of the possible causes underlying the functional segregation of the dorsal striatum. In this direction, it has been shown that the upward and downward transitions slopes of cortical neurons are governing by GABA inhibition (Craig et al., 2013; Mann et al., 2009; Perez-Zabalza et al., 2020; Sanchez-Vives et al., 2010).

Please, note that this question is also discussed answering reviewer 2.

We have now modified the text relative to the DH ratio, in the Materials and methods, Results and Discussion sections, to clarify them.

5) The method used to do the delineation of the DCS is not clear and should be better explained in the main text.

We thank the reviewer for this assessment. We have extended the section “DLS and DMS are two non-overlapping functional circuits in mouse” in order to clarify and provide further context to the study of the possible boundary between circuits in the DCS coordinates.

In brief, we used the SVM that had been used previously to classify these two populations to study the distribution of DCS-MSN in the parameter space relative to the populations of DLS- and DMS-MSNs. The trained SVM contains a hypervector that maximises the separation between the two populations; in other words, it let us project our data into a new axis which has been optimised to distinguish between DLS and DMS MSNs. In this space, DLS- and DMS-MSNs have a known distribution and because of these, we could introduce a new pool of MSNs (recorded in DCS) and compare it with the previous ones. This let understand whether the DCS-MSNs created a continuum between the DLS- and DMS-MSNs populations along the decision axis or whether this separation remained after the addition of the new pool of MSNs, depending on the distribution of the population of DCS-MSNs along this axis.

6) Slow wave oscillations are a specific regime within these networks and even though informative, analyzing the networks in that state might be limiting. Mahon et al., 2006, they show a very different activity in the corticostriatal oscillations between SWO and awake animals. The results of the present study should be discussed in light of the different regimes existing in the corticostriatal pathways depending on the vigilance state of the animals.

We have extended the Discussion regarding the validity of our study when compared to awake states in the context of Mahon et al., 2006.

First, the use of the SWO state let us benefit from a well-known brain state, on which cortically generated slow oscillations will propagate along the cortex and to the striatum, with well-defined spatial-temporal properties. This scenario is ideal to understand cortico-striatal communication, as we can study the propagation of each wave to the striatum as it travels through the cortex. In addition, it let us understand how these oscillations are integrated simultaneously in DLS and DMS thanks to the double patch-clamp recordings. These particular experiments would not have been possible without the mechanical stability that results from an anaesthetised preparation.

Second, we have discussed the work by Mahon et al., 2006. The authors described that, similar to what it happens in mice or human cortex, MSNs change from a highly synchronous SWO state during anaesthesia/sleep to an asynchronous state during awake states. Previous works (Getting, 1989; Luczak et al., 2015) have shown that the functional constrains that will shape the activity of a common circuits are shared between spontaneous and evoked states. Therefore, our results discussing the differences in the striatal organization provide insights about the functional organization of the striatum during different brain and behavioural states.

Related to this, previous works have already demonstrated that the study of the SWO is useful to understand human and mice neurological disorders (Amiri et al., 2019; Busche et al., 2015; Ruiz-Mejias et al., 2016). The SWO has been considered as the default activity pattern of the cortex and the cortex is the major source of inputs to the striatum. One of the main functions of the slow-wave sleep is to preserve the proper connectivity within the circuits (Sanchez-Vives et al., 2017) and has been extensible related with memory consolidation (Marshall et al., 2006), thus from this point of view, it makes sense to extract information of the circuits based on the analysis of the SWO. In fact, the SWO was previously suggested as a model to study functional connectivity between brain circuits (Massimini et al., 2004; Murphy et al., 2009).

7) One last suggestion is that direct and indirect pathway data could be presented at the beginning with intrinsic properties and up states characterization of the MSNs in DMS and DLS to avoid coming back to intrinsic properties at the end of the manuscript.

We thank reviewer #1 for this suggestion. We have considered the reordering of the narrative following this indication. While the suggested order will unify the discussion about the intrinsic properties between regions and MSN types, we think that the current organization puts the focus on the main point of the paper, which is the differentiation between DLS and DMS. Therefore, we think it is important to maintain the present order, going from the general question to the specific aspects, including subpopulation of neurons as direct or indirect MSNs. But, if the reviewer #1 still considers that this is an essential point to understand the study, we will reorganize our manuscript.

Reviewer #2:This interesting study presents a number of analyses of in vivo intracellular recordings of the membrane potential activity of MSNs in the striatum, which indicate that there are differences in electrophysiological properties in dorsomedial vs dorsolateral striatum. Furthermore, there appears to be a discrete demarcation between neurons in each area with only a small overlap and no gradient. The authors then go on to record from afferent areas to determine if the patterns of activity in the MSNs are echoed by differences in neuronal activity in functional areas projecting to the striatum. Finally, they further demarcate the neurons in each area by whether they are dMSNs or iMSNs, and find that there were further differences of neurons in the lateral striatum only.Although I can understand how temporal measures such a durations of states and trajectory of transitions can be quantified using the NA-MEMD algorithm, I cannot see how the actual average and SD of the up and down state membrane potentials have been extracted from the IMFs, when looking at those exemplified in Figure 2—figure supplement 2C and the SWO also shown in Figure 1C. Was there an additional step needed to calculate a probability density function from the distribution of the membrane potentials, as in Figure 5C? If not, because as the authors say it is the first time that the algorithm has been used to analyse membrane oscillations of single neurons, a much more detailed methodological graphical explanation as to how the potentials have been derived is warranted.

We thank Dr Reynolds for his supportive comments. We have clarified the use of the NA-MEMD in the new manuscript, as it was sometimes unclear whether a certain analysis had been done using the original raw trace or a certain IMF. The results shown in Figures 1,4 and 7 were computed using the original traces, and the Up states were isolated using a standard moving threshold, as described in the Materials and methods section.

The authors found "an Up state in FrA preceding an Up state in V1 in the 71% of times (p value< 0.0001), data not shown" – this is itself very interesting and the data SHOULD be shown as further suggestive (correlative) evidence that the propagation of Up state timing from lateral to medial striatum is driven by frontal to visual cortex transition of slow wave.

We thank Dr Reynolds for this suggestion. We have included a new panel (panel F) in Figure 6 to show this data.

The observation of a difference between cellular properties and SWO measures in lateral and medial regions is just that, without any expose of the underlying reason. This paper could be much more enriched by some investigation of a mechanism that might explain these differences. In lieu of a full morphological comparison, one possibility the authors do raise that is quite achievable in a relatively short time is considering the effects of differences in dopamine innervation, either directly or indirectly via cholinergic neuronal innervation. Indeed there is known to be medial to lateral gradients of many neuromodulatory components including dopamine receptors which might contribute to the regional differences. The authors should consider undertaking a series of comparison experiments under blockade of D1 and D2 antagonists (separately and together, or 6-OHDA depletion better still for the latter) and see if this normalises the observed differences between neurons. This is relatively more straightforward than trying to normalise FSI innervation and in my opinion is much more likely to provide at least a partial explanation.

We thank Dr Reynolds for his suggestion.

We agree with the hypothesis that dopaminergic innervation could play a direct or indirect role on the functional discrimination of the striatum, together with other mechanisms such as the corticostriatal innervation and local inhibition. However, we consider that the impact of DA on the striatal SWO is beyond the scope of this study. Our classifier is created on the base of 197 striatal neurons recorded from the intact circuit in a heathy condition.

Dopamine depletion induces changes in the striatal microcircuits but also in the cortical activity. Using in vivo patch-clamp recording in anesthetized mouse, the same technique used in our study, it was shown that the Up state frequency of single cortical neurons recorded in S1 decreases after 6-OHDA lesion (Ketzef et al., 2017). Hence, dopamine depletion will alter the corticostriatal transmission presynaptically, in the cerebral cortex, inducing unspecific changes in addition to the striatal ones, which can mask the mechanism underlying the striatal division. On the other hand, these experiments would only provide a partial explanation. Striatal neurons express different types of DA receptors, in MSNs and in different types of interneurons. In addition, DA receptors are expressed presynaptically at the glutamatergic terminals. Therefore, the absence of DA will not provide the precise mechanism underlying the possible changes on MSNs Up states.

In the new version of the manuscript, we have emphasized the study of the properties helping to discern between DLS and DMS. Figure 4 describe the features selected by our classifier that are relevant to discriminate between striatal regions.

Because it is known that corticostriatal axons are characterized by some degree of anatomical specificity, our first hypothesis was to explore whether cortical inputs are relevant to discriminate between the DLS and DMS. To answer this hypothesis, whole-cell recordings were performed in cortical neurons from FrA, M1, S1 and V1. We found that the number of peaks in the up states were clearly higher in M1 and S1 as well as in the DLS-MSNs. Thus, the structure of these complex fluctuations of the membrane potential of cortical Up states from M1 and S1 neurons are reproduced in the DLS-MSNs but not in the DMS-MSNs. This result, together with the previous anatomical description, strongly suggest that M1 and S1 inputs are sculpting the Up states of DLS-MSNs, in which the excitatory glutamatergic inputs are directly transmitted to MSNs, while the inhibitory ones must be mediated by one or several types of interneurons. Therefore, we conclude that cortical inputs are important for the functional division between the DLS and DMS.

In addition to the result regarding the number of peaks, the information about the Up to Down states slopes between DLS- and DMS-MSNs rises the classification accuracy up to ~90%. To illustrate the divergence of upward and downward transition slopes we have included a new plot (new Figure 4 panel F). It is known that GABA release modifies Up states slopes in cortical neurons. This effect is mediated by the activation of postsynaptic GABAA (Sanchez-Vives et al., 2010) and GABAB receptors (Craig et al., 2013; Mann et al., 2009; Perez-Zabalza et al., 2020) for Down to Up and Up to Down transition slopes respectively. Because MSNs display IPSPs during spontaneous Up states (Reig and Silberberg, 2014) and it has been described a different number of PV and activity of cholinergic interneurons along dorsal striatum, we discuss that the observed Up states slopes divergence between DLS- and DMS-MSNs can be mediated by differences on inhibition. In order to explore this possibility, we analysed the time course of the depolarized and hyperpolarized events during the Up states (Figure 4D), and then we calculated its average ratio (Figure 4E). The result shows a higher positive and negative ratio during the Upward and Downward transition slope respectively for DMS-MSNs. This result together with the previous descriptions of interneurons distribution along dorsal striatum suggest that local inhibition contribute to the division of the DLS and DMS circuits.